# Hippocampal interneuronal dysfunction and hyperexcitability in a porcine model of concussion

Alexandra V. Ulyanova [1,2], Christopher D. Adam[1], Carlo Cottone[1], Nikhil Maheshwari[1], Michael R. Grovola[1], Oceane E. Fruchet[1], Jami Alamar[1], Paul F. Koch[1], Victoria E. Johnson[1], D. Kacy Cullen [1,2] & John A. Wolf [1,2 ✉]

Cognitive impairment is a common symptom following mild traumatic brain injury (mTBI or concussion) and can persist for years in some individuals. Hippocampal slice preparations following closed-head, rotational acceleration injury in swine have previously demonstrated reduced axonal function and hippocampal circuitry disruption. However, electrophysiological changes in hippocampal neurons and their subtypes in a large animal mTBI model have not been examined. Using in vivo electrophysiology techniques, we examined laminar oscillatory field potentials and single unit activity in the hippocampal network 7 days post-injury in anesthetized minipigs. Concussion altered the electrophysiological properties of pyramidal cells and interneurons differently in area CA1. While the firing rate, spike width and amplitude of CA1 interneurons were significantly decreased post-mTBI, these parameters were unchanged in CA1 pyramidal neurons. In addition, CA1 pyramidal neurons in TBI animals were less entrained to hippocampal gamma (40–80 Hz) oscillations. Stimulation of the Schaffer collaterals also revealed hyperexcitability across the CA1 lamina post-mTBI. Computational simulations suggest that reported changes in interneuronal physiology may be due to alterations in voltage-gated sodium channels. These data demonstrate that a single concussion can lead to significant neuronal and circuit level changes in the hippocampus, which may contribute to cognitive dysfunction following mTBI.

[1] Center for Brain Injury and Repair, Department of Neurosurgery, University of Pennsylvania, Philadelphia, USA. [2] Center for Neurotrauma, Neurodegeneration, and Restoration, Corporal Michael J. Crescenz Veterans Affairs Medical Center, Philadelphia, USA. ✉email: wolfjo@pennmedicine.upenn.edu

Concussion, or mild TBI (mTBI), is a prevalent form of brain injury, affecting approximately 1.3 million people every year in the US alone[1]. Cognitive impairment is common following concussion and deficits in episodic memory, learning, and delayed recall have been described[2–4]. Importantly, a growing body of data suggests that neurocognitive dysfunction observed post-concussion can persist for years after initial insult in some people[5–9]. Given the important role of the hippocampus in learning and memory, it is a central region of interest in diseases affecting cognitive processes[10]. While the underlying mechanisms of cognitive dysfunction following mTBI remain largely unknown, structural changes in the human hippocampus up to 1-year post-concussion have been reported using imaging techniques[11,12]. Recent reports also link concussion to hyperexcitability[13,14]. Moreover, recent reports based on the epidemiological data may indicate a 2-times increased risk for post-traumatic epilepsy in adult population with mTBI[15–17].

Rodent studies of TBI have shown changes in neuronal firing and local hippocampal oscillations, potentially suggesting abnormalities in temporal coding, a crucial element of episodic memory formation and encoding[18–20]. While studies utilizing rodent models of TBI can provide some mechanistic insight into electrophysiological changes following TBI, brains of large animals such as pigs, with their gyrencephalic structure and appropriate white-to-grey matter ratios, more closely resemble human architecture and are important for an accurate biomechanical modeling of all aspects of human TBI[21]. Using a biomechanically relevant and gyrencephalic model of concussion in swine, we demonstrated injury-related compensatory mechanisms and a reduction in axonal function in the ex vivo hippocampus following TBI in the apparent absence of intra-hippocampal neuronal or axonal degeneration[22]. Deafferentation of temporal lobe structures (such as the hippocampus) via axonal pathology has been proposed as a driver of hyperexcitability in this model and may disrupt the precise timing of hippocampal oscillations and associated neuronal activity[23]. Diffuse axonal injury is the predominant pathology observed in this preclinical model, however direct evidence that concussion disrupts neuronal network activity, specific neuronal subtypes, or laminar oscillations in the hippocampus of the intact swine has yet to be demonstrated.

We utilized an established porcine model of closed-head, rotational acceleration to induce concussion, and then applied high-density in vivo electrophysiology techniques to examine neuronal and oscillatory changes in the intact hippocampus at 7 days post-injury. Notably, the function of hippocampal circuits was assessed using laminar, multichannel silicon probes specifically designed for recording from deep brain structures of large animals[24,25]. Laminar oscillatory field potentials and single unit activity were examined and compared between injured and control animals under isoflurane anesthesia. Our data suggest that even a single concussion may alter both individual neuronal function and circuit-level activity in the hippocampus and may provide insight into aspects of hyperexcitability and cognitive dysfunction following concussion.

## Results

**Concussion affects electrophysiological properties of hippocampal CA1 neurons**. To examine changes in hippocampal circuitry induced by concussion, in vivo electrophysiological recordings were obtained using 32-channel silicon probes that spanned the laminar structure of CA1 in the dorsal hippocampus (Fig. 1a). These probes allowed recording of local field potential across layers of CA1 and isolation of single units using available spike sorting algorithms (see Methods)[26]. Electrophysiological

properties of CA1 neurons such as firing rate, spike width and amplitude (Fig. 1b) were calculated and compared between control ($n_{animals} = 8$; $n_{cells} = 125$) and post-mTBI ($n_{animals} = 9$; $n_{cells} = 93$) animals at 7 days post-concussion (Fig. 1c). While the number of recorded cells per animal was not different between the groups (control $= 16 \pm 3$ cells vs. post-mTBI $= 10 \pm 3$ cells, $p = 0.2397$, see Supplementary Fig. 1a), the firing rate, spike width, and spike amplitude were all significantly reduced in post-mTBI animals (Fig. 1c). The firing rate of $4.43 \pm 0.50$ Hz in the control animals was reduced to $2.09 \pm 0.27$ Hz in the post-mTBI animals ($p = 0.0002$). The spike width of $0.358 \pm 0.015$ m in the control group was significantly reduced to $0.270 \pm 0.008$ ms in the post-mTBI group ($p < 0.0001$), and the spike amplitude was significantly reduced from $300 \pm 27\,\mu V$ in the control group to $187 \pm 27\,\mu V$ in the post-mTBI group of animals ($p = 0.0003$). These findings suggest that concussion may alter the intrinsic properties of CA1 neurons, which could affect their synchronization with hippocampal oscillations.

**Pyramidal neurons in CA1 preserve their firing properties following concussion**. To further characterize the effects of mTBI on the hippocampal circuitry and to assess changes in specific subpopulations of cells in CA1, recorded cells were separated into putative CA1 pyramidal cells or putative CA1 interneurons[27]. Previously published criteria for rodent hippocampal cells were used for classification[18,28], with several modifications accounting for species differences. Briefly, the probe's anatomical location coupled with electrophysiological characteristics were utilized to exclude cells recorded at the bottom portion of the probe, as these cells could be identified as putative dentate granule cells, mossy cells, or interneurons in the dentate (control: $n_{cells} = 14$, post-mTBI: $n_{cells} = 5$). Additionally, unclassifiable cells located throughout the length of the probe (control: $n_{cells} = 10$, post-mTBI: $n_{cells} = 16$) were omitted from the analysis. Next, electrophysiological parameters (firing rate, spike width, and first moment of autocorrelogram, Figs. 1b, 2a, 3a) of the remaining single units were used to classify the remaining neurons (see Methods and Supplementary Fig. 1d for more details) as either putative CA1 pyramidal cells (control: $n_{pyr} = 49$, post-mTBI: $n_{pyr} = 37$; Fig. 2) or putative CA1 interneurons (control: $n_{int} = 52$, post-mTBI: $n_{int} = 38$; Fig. 3). The number of either pyramidal cells (control $= 7 \pm 2$ cells vs. post-mTBI $= 4 \pm 1$ cells) or interneurons (control $= 6 \pm 2$ cells vs. post-mTBI $= 4 \pm 2$ cells) per animal was not different between the control and post-mTBI groups (CA1 pyramidal cells: $p = 0.3515$; CA1 interneurons: $p = 0.4314$).

Electrophysiological properties of putative CA1 pyramidal cells in the post-mTBI group such as firing rate (control $= 2.27 \pm 0.25$ Hz vs. post-mTBI $= 1.70 \pm 0.27$ Hz; $p = 0.0555$), spike width (control $= 0.336 \pm 0.022$ ms vs. post-mTBI $= 0.280 \pm 0.014$ ms; $p = 0.2269$), and spike amplitude (control $= 256 \pm 36\,\mu V$ vs. post-mTBI $= 238 \pm 50\,\mu V$; $p = 0.1372$) were not significantly different between the control ($n = 8$) and post-mTBI ($n = 9$) groups of animals at 7 days following concussion (Fig. 2b). Recent studies in rodent models of TBI have shown changes in both neuronal firing and hippocampal local field potentials, suggesting abnormalities in temporal coding, a process through which tightly coordinated network activity seen in the local field potential controls the precise timing of neuronal firing[18–20]. Therefore, to examine whether CA1 pyramidal cells continue to interact with hippocampal oscillations properly following concussion, entrainment of CA1 pyramidal cells to local field potentials was also investigated (Fig. 2c). Notably, entrainment strength (a measure that determines how tightly coupled neuronal firing is to particular phases of an oscillation) of CA1 pyramidal cells to local oscillations was significantly reduced (Fig. 2d) in the

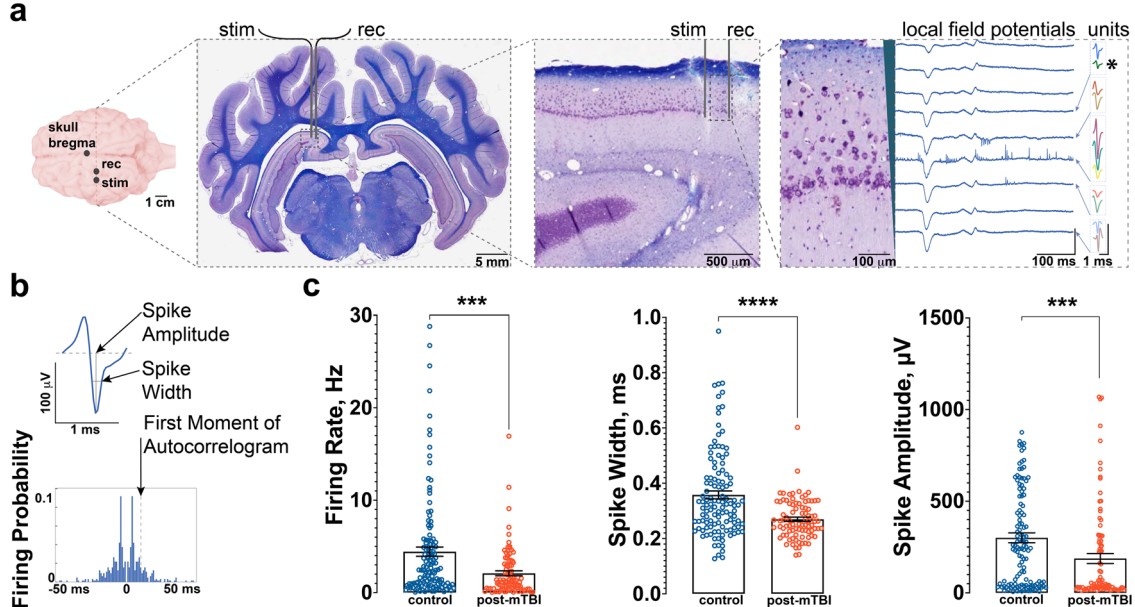

**Fig. 1 Concussion affects electrophysiological properties of hippocampal CA1 neurons. a** Anatomical locations for recording and stimulation electrodes depicted relative to bregma point (scale bar = 1 cm). A representative coronal section through the plane of recording/stimulation showing hippocampal anatomy, stained with LFB/CV as previously described (scale bar = 5 mm)[25]. Stimulation electrode and recording electrode shown overlaid on dorsal hippocampus (scale bar = 500 μm). Local field potentials (scale bars: x axis = 100 ms, y axis = 1 mV) and multiple single units (scale bars: x axis = 1 ms, y axis = 300 μV) recorded with 32-channel laminar probe (scale bar = 100 μm). Note multiple single units present on the same channel (boxed and color coded). **b** A representative single unit (green, depicted with * in **a**), displaying electrophysiological parameters such as firing rate, spike width, and spike amplitude. First moment of autocorrelogram was also calculated for each spike based on firing probability. **c** At 7 days post-mTBI, the firing rate of hippocampal CA1 neurons was significantly reduced in the control (4.43 ± 0.50 Hz) vs. post-mTBI (2.09 ± 0.27 Hz) groups of animals ($p = 0.0002$). In addition, the spike width of 0.358 ± 0.015 ms recorded in the control animals was significantly reduced to 0.270 ± 0.008 ms in the post-mTBI animals ($p < 0.0001$). The spike amplitude was also significantly reduced from 300 ± 27 μV in the control animals to 187 ± 27 μV in the post-mTBI animals ($p = 0.0003$). Data presented as mean ± SEM.

62–82 Hz frequency band (control = 0.020 ± 0.004 vs. post-mTBI = 0.005 ± 0.001, $p = 0.0036$) and in the 103–117 Hz frequency band (control = 0.018 ± 0.005 vs. post-mTBI = 0.003 ± 0.001, $p = 0.0007$). Additionally, we found a significant reduction in gamma oscillation (35–75 Hz) power (a measure that determines the strength of a particular oscillation in the overall EEG signal) detected locally to pyramidal CA1 cells (Fig. 2e, $p < 0.01$). These results indicate that concussion disrupts the generation of gamma oscillations in the pyramidal cell layer due to the changes in afferent inputs coming into CA1 and/or alterations in local CA1 pyramidal-interneuron or interneuron-interneuron interactions.

**Inhibitory CA1 interneurons are preferentially affected following concussion**. Analysis of putative CA1 interneurons revealed that their electrophysiological properties were significantly altered. In contrast to CA1 pyramidal cells, CA1 interneurons had a significantly reduced firing rate (control = 7.46 ± 1.0 Hz vs. post-mTBI = 3.09 ± 0.55 Hz, $p = 0.0008$), spike width (control = 0.373 ± 0.022 ms vs. post-mTBI = 0.263 ± 0.009 msec, $p = 0.0004$), and spike amplitude (control = 367 ± 50 μV vs. post-mTBI = 170 ± 43 μV, $p = 0.0036$) at 7 days post-concussion (Fig. 3b). Interestingly, the firing rate of CA1 interneurons decreased as rotational velocity increased, changing significantly from 7.46 ± 1.0 Hz in control animals ($n_{animals} = 8$, $n_{cells} = 52$) to 3.78 ± 0.93 Hz in animals injured at ~190 rad/sec ($n_{animals} = 4$, $n_{cells} = 20$), and further to 2.33 ± 0.51 Hz in animals injured at ~260 rad/sec ($n_{animals} = 5$, $n_{cells} = 18$) ($p = 0.0030$, Supplementary Fig. 1b). Since the electrophysiological properties of CA1 pyramidal cells were not substantially altered (Fig. 2b), these results indicate that CA1 interneurons are preferentially

affected at 7 days following a single inertial injury. Notably, entrainment of CA1 interneurons to oscillations in the 4–6 Hz frequency range was significantly increased 7 days post-concussion (control = 0.013 ± 0.005 vs. post-mTBI = 0.041 ± 0.014, $p = 0.0415$, Fig. 3c, d). Additionally, there was a significant reduction in the power of high frequency oscillations (frequency range = 80–300 Hz) detected locally to interneurons (but not pyramidal cells) in CA1 cells, potentially due to the observed reduction in the firing rate of CA1 interneurons or desynchronization of the CA1 network (Fig. 3e, $p < 0.01$). These results indicate that concussion alters intrinsic and/or synaptic properties of CA1 interneurons, which affects local gamma and high-frequency oscillations presumably generated by local CA1 interneurons. These changes likely alter the way CA1 pyramidal cells interact with local hippocampal oscillations.

**Hippocampal hyperexcitability post-concussion**. Reduced excitability of GABAergic inhibitory interneurons without a corresponding change in the activity of excitatory neurons in the hippocampus has been previously linked to hippocampal hyperexcitability[29–32]. Moreover, the differential response to stimulation ex vivo has been previously reported between control and post-mTBI animals using this model[22], further suggesting hyperexcitability in hippocampus post-concussion. Here, we used electrical stimulation of the Schaffer collateral inputs coming into the CA1 layer from CA3 in a subset of animals (control = 3, post-mTBI = 7) to assess changes in the excitability levels in the intact brain[25]. Representative examples of the evoked laminar field potentials in dorsal CA1 using Schaffer collateral stimulation are shown as averaged responses over a 300 ms time period

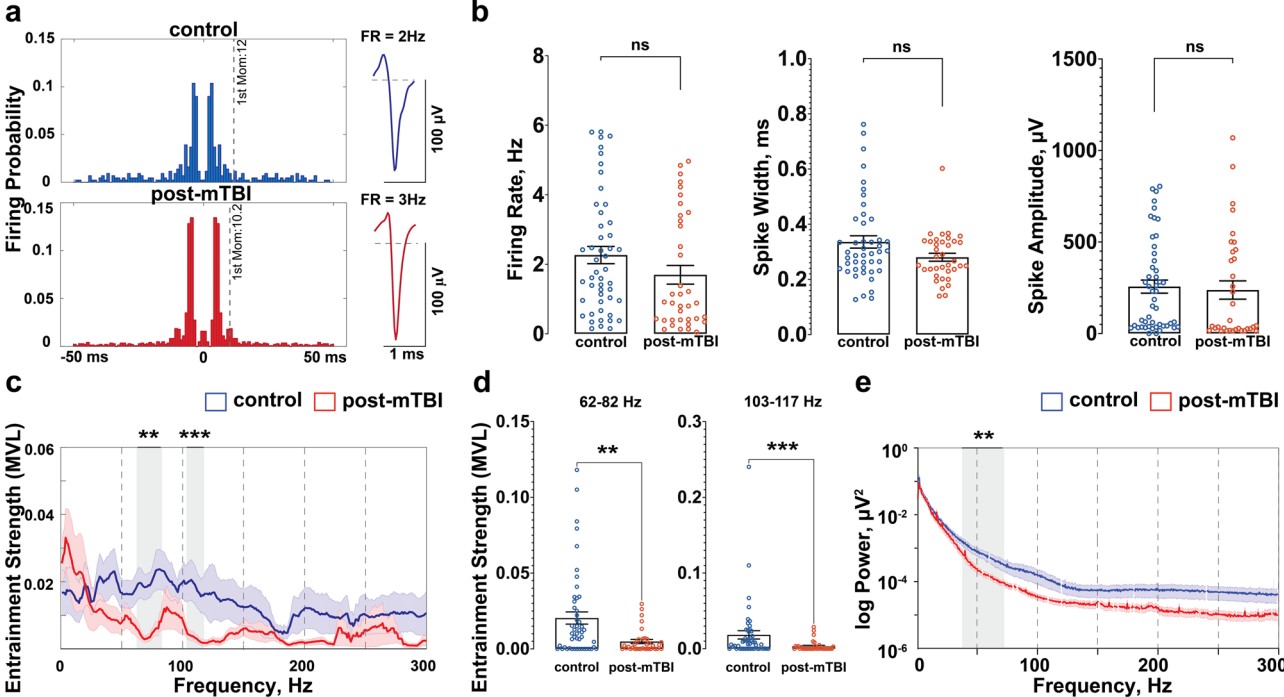

**Fig. 2 Pyramidal neurons in CA1 preserve their firing properties following concussion but are less entrained to gamma. a** Representative examples of putative CA1 pyramidal cells from the control (blue) and post-mTBI (red) animals are shown. **b** Electrophysiological properties of pyramidal cells were not changed 7 days following mTBI (control: $n_{pyr} = 49$, post-mTBI: $n_{pyr} = 37$). The parameters analyzed include firing rate (control = 2.27 ± 0.25 Hz, vs. post-mTBI = 1.70 ± 0.27 Hz; $p = 0.0555$), spike width (control = 0.336 ± 0.022 ms vs. post-mTBI = 0.280 ± 0.014 ms; $p = 0.2269$), and spike amplitude (control = 256 ± 36 μV vs. post-mTBI = 238 ± 50 μV; $p = 0.1372$). Data presented as mean ± SEM. **c** Entrainment of CA1 pyramidal cells to local hippocampal oscillations was calculated as a mean vector length (MVL) and compared between the control and post-mTBI groups in the 0–300 Hz frequency range. Data presented as mean ± SEM. **d** Entrainment of CA1 pyramidal cells was significantly decreased in the 62–82 Hz (control = 0.020 ± 0.004 vs. post-mTBI = 0.005 ± 0.001, $p = 0.0036$) and the 103–117 Hz (control = 0. 018 ± 0.005 vs. post-mTBI = 0.003 ± 0.001, $p = 0.0007$) frequency ranges in the post-mTBI animals. Data presented as mean ± SEM. **e** Power spectral density analysis shows a significant decrease in the 35–75 Hz frequency range ($p < 0.01$) following mTBI. Data presented as mean ± STD. Noise associated with 60 Hz cycle was removed manually.

(20 stimulations, 2 s apart) for a control (Fig. 4a, blue traces) and a post-mTBI (Fig. 4a, red traces) animals. During a series of the paired-pulse (PP) stimulations in the Shaffer collaterals (inter-pulse interval of 30 ms, 20 repeats, 5 s apart), a sustained depolarization over 1 s long, represented by negative deflection of the extracellular local field potentials[33], was observed in a subset ($n = 2$ out of 5) animals injured at ~260 rad/sec (Fig. 4b, panel 1). Notably, this depolarizing shift (Fig. 4c) was detected across all hippocampal layers and has features similar to the paroxysmal depolarizing shift previously observed in epileptic brain[34]. In the example shown in Fig. 4c, the depolarizing shift happened approximately 4 s after the third PP stimulation and continued for over 1 s, overlapping with the fourth PP stimulation. In addition, depolarization was followed by a loss of synaptic responses to PP stimulation (Fig. 4b, panel 2). Other forms of hippocampal hyperexcitability such as paroxysmal rhythmic spikes were observed in the form of synchronized activity in the 6–8 Hz frequency range across all laminar layers in injured but not control animals, potentially suggesting increased hyperexcitability and synchronization of hippocampal circuitry prior to the introduction of a stimulus (Fig. 4d).

**Model of Parvalbumin-positive (PV⁺) CA1 interneurons reveals potential dysfunction of sodium channels.** Interneurons in the hippocampus are involved in the generation of gamma oscillations and provide input-specific inhibition to pyramidal cells via feedforward and feedback inhibition[33,35,36]. Specifically, it has been previously shown that parvalbumin-positive (PV⁺) basket cells provide peri-somatic, feedforward inhibition in the

hippocampal CA1 region[37,38]. These cells stabilize hippocampal circuitry during changes in input and control the selection of cell ensembles by allowing only certain number of pyramidal cells to fire in response to the stimulus[33,39,40]. Interestingly, pathological changes in PV⁺ interneurons in CA1 have been also shown to play a crucial role in the development of epilepsy[41,42]. To elucidate potential mechanisms underlying the electrophysiological changes observed in CA1 interneurons in our animal model of concussion, we used the NEURON simulation environment[43,44], specifically a model of hippocampal PV⁺ basket cells described previously[45], and adjusted parameters to fit the observed decrease in spike size and firing rate observed in interneurons of mTBI animals (Table 1). The full individual parameter spaces of the CA1 interneuronal model were altered by hand without an a priori hypothesis in both the soma and dendrites to examine whether changing multiple parameters might fit the reduced firing rate, spike width and spike amplitude observed in CA1 interneurons post-mTBI (Fig. 5). The best fit between model-generated values (Fig. 5a, green) and experimental data of injured CA1 interneurons (Fig. 5a, red) was achieved by changing the inactivation of voltage-gated sodium channels ($h_{inf}$) in the model (Table 1). The best fit for the PV⁺ interneuron model corresponds to a decrease in firing rate (model = 60% vs. experimental data = 59%), spike width (model = 28% vs. experimental data = 29%), and spike amplitude (model = 27% vs. experimental data = 54%) (Fig. 5b, c; control - blue trace, post-mTBI - red trace, best-fit - green trace).

As detailed in Table 1, the experimental data is best described by a negative shift of $h_{inf}$ towards more hyperpolarized voltages,

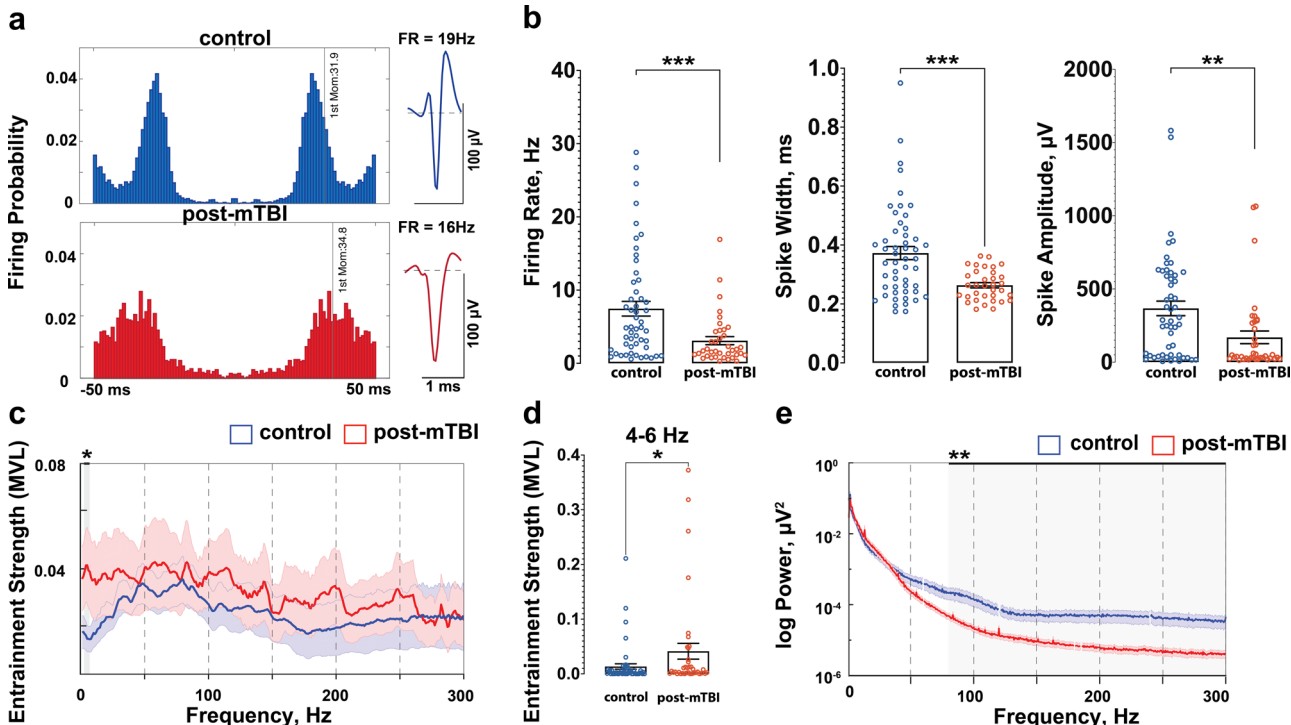

**Fig. 3 Interneurons in CA1 are preferentially affected following concussion. a** Representative examples of CA1 interneurons from control (blue) and post-mTBI (red) animals. **b** Electrophysiological properties of CA1 interneurons were significantly altered 7 days post inertial injury. Firing rate of CA1 interneurons was significantly reduced from 7.46 ± 1.0 Hz in control animals (blue) to 3.09 ± 0.55 Hz (red) in post-mTBI animals (control: $n_{animals} = 8$, $n_{cells} = 52$; post-mTBI: $n_{animals} = 9$, $n_{cells} = 38$; $p = 0.0008$). The spike width was significantly reduced from 0.373 ± 0.022 ms in control animals (blue) to 0.263 ± 0.009 ms in post-mTBI animals (red) ($p = 0.0004$). Spike amplitude was also significantly reduced from 367 ± 50 μV in control animals (blue) to 170 ± 43 μV in post-mTBI animals ($p = 0.0036$). Data presented as mean ± SEM. **c** Entrainment of interneurons in CA1 to local hippocampal oscillations was calculated as a mean vector length (MVL) and compared between the control and post-mTBI groups in the 0–300 Hz frequency range. Data presented as mean ± SEM. **d** Entrainment of interneurons in CA1 was significantly increased in the 4–6 Hz frequency range (control = 0.013 ± 0.005 vs. post-mTBI = 0.041 ± 0.014, $p = 0.0415$). Data presented as mean ± SEM. **e** Power of high frequency oscillations was significantly decreased in the 80–300 Hz frequency range ($p < 0.01$). Data presented as mean ± STD. Noise associated with 60 Hz cycle was removed manually.

with the midpoint of the steady state inactivation curve changing from −35 mV to −44.5 mV (Fig. 5b, arrow). Under these modeling conditions, the window current of voltage-gated sodium channels, defined as an overlap between the activation ($m_{inf}$) and inactivation ($h_{inf}$) curves (calculated as area under the curves), is decreased by ~30% following inertial injury. The model output using the best fit parameters is shown in Fig. 5c (control – blue, post-mTBI – red, best fit model – green). These results suggest that pathological changes in the inactivation of voltage-gated sodium channels may be a potential mechanism for $PV^+$ interneuron dysfunction post-mTBI.

## Discussion

Concussion (or mild TBI) is a prevalent type of brain injury with the potential for long-term consequences[46,47]. Specifically, concussion is known to lead to acute and chronic cognitive deficits in a subset of mTBI patients, which manifest as poor performance in tests of memory, learning, and delayed recall up to several weeks post injury[2–4]. The hippocampus is involved in learning and memory processes[10], and its role in integrating information to support these processes may be particularly sensitive to temporal coding disruptions associated with axonal injury or disruptions in oscillatory activity[23]. Temporal coding in the CA1 region of the hippocampus supports the acquisition, consolidation, and retrieval of spatial and episodic memories by integrating information from entorhinal and CA3 inputs in a temporally precise manner before routing it to downstream structures[6,11,48,49]. Changes in neuronal firing and local hippocampal oscillations

have been reported in rodent models of mTBI, suggesting abnormalities in temporal coding[18–20]. However, studies investigating electrophysiological changes following clinical concussion are sparse due to the lack of data from human patients[50].

Large animal gyrencephalic models offer several advantages in modeling the complex biomechanics and resultant pathophysiology of inertial injury[51], and these models may be the most relevant for understanding the effects of concussion on cognitive processing and memory. We have previously reported changes in hippocampal circuitry following mTBI using an ex vivo slice preparation[22]. In the current study, we advanced our investigation of electrophysiological changes in CA1 physiology following rotational acceleration injury by performing high-density laminar electrophysiology in the intact anesthetized swine at 7 days post-mTBI. We found that CA1 interneurons had a significantly decreased firing rate, spike amplitude, and spike width, in the injured compared to the control animals. Moreover, we found that concussion resulted in hippocampal hyperexcitability in vivo, which confirms findings from prior work utilizing this model[22]. Because CA1 activity is known to support learning and memory processes, our combined observations suggest that mTBI-associated changes in CA1 physiology may underlie some aspects of cognitive disruption following mTBI.

We found that concussion-induced changes in neuronal firing in CA1 were cell-type specific and preferentially affected CA1 interneurons. We showed that concussion, induced by inertial rotation, led to changes in the firing rate and spike shape of CA1 interneurons but not pyramidal cells. Specifically, CA1

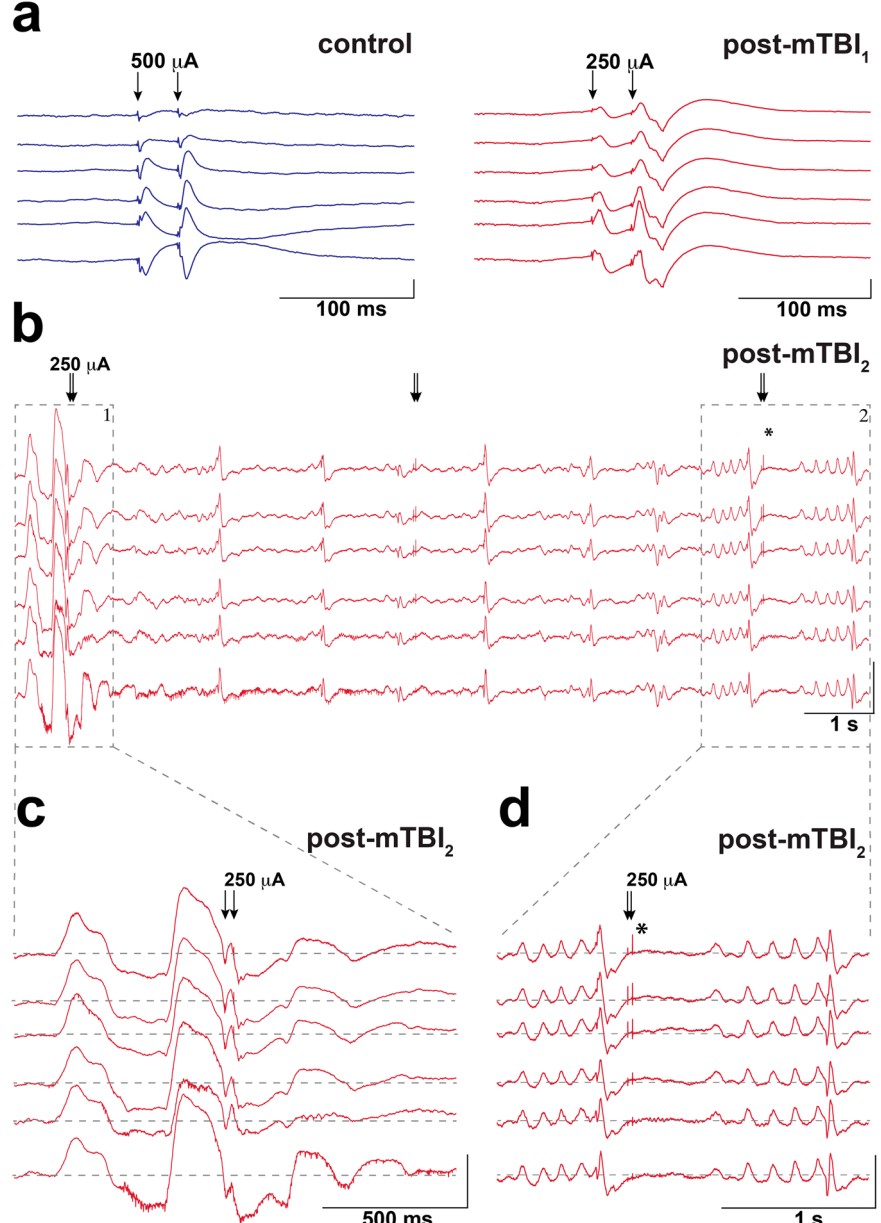

**Fig. 4 Hippocampal hyperexcitability following concussion. a** Averaged CA1 laminar field responses to a train of paired pulse (PP) stimulation (inter-pulse interval, IPI = 50 ms, 2 s in between stimulations, 20 stimulations total) in Schaffer Collaterals (arrows) are shown for a control (stimulation amplitude = 500 µA) and post-mTBI (stimulation amplitude = 250 µA) animals (scale bars: x axis = 100 ms, y axis = 1 mV). **b** A depolarizing shift was observed in a subset (n = 2) of post-mTBI (but not control) animals during PP stimulation (IPI = 30 ms, stimulation amplitude = 250 µA, 5 s in between stimulations, 20 stimulations total). In this example, the depolarizing shift happened approximately 4 s following an application of 3rd PP stimulation and continued for over 1 s (panel 1), overlapping with the 4th PP stimulation (arrows). A prolonged depolarization followed by a loss of synaptic responses to PP stimulation (arrows) was also observed in all layers of hippocampus (panel 2, asterisk) (scale bars: x axis = 1 s, y axis = 0.5 mV). **c** Magnified view of panel 1 in (**b**) shows a period of prolonged depolarization as negative deflection over baseline, with multi-unit activity present on some channels (scale bars: x axis = 500 ms, y axis = 0.5 mV). **d** Magnified view of panel 2 in (**b**) shows other forms of hippocampal hyperexcitability such as a loss of synaptic responses stimulation (asterisk) and paroxysmal rhythmic spikes in the form of synchronized activity in the 6–8 Hz frequency range across all laminar layers (scale bars: x axis = 1 s, y axis = 0.5 mV).

interneurons had decreased firing rates at 7 days following concussion, potentially contributing to a decrease in local inhibition resulting in increased excitation and hyperexcitability in the CA1 pyramidal cell layer. Our NEURON modeling results indicated that the decreased firing rate and a reduction in spike amplitude and width observed in CA1 interneurons of mTBI animals may be due to a negative shift in the inactivation of voltage-gated Na$^+$ channels. While other groups have proposed a coupled left-shift of activation/availability of sodium channels using an in vitro

model of stretch injury[52], our computational model did not predict this negative shift in sodium channel activation but instead predicted a negative shift in sodium channel inactivation. Notably, pathological changes affecting the inactivation of voltage-gated Na$^+$ channels have been previously reported following injury[53,54]. Using an in vitro model of axonal stretch injury, we previously demonstrated that traumatic deformation of axons induced abnormal sodium influx through TTX-sensitive Na$^+$ channels, leading to an increase in the intracellular Ca$^{2+}$

**Table 1 NEURON-simulation environment model of parvalbumin-positive (PV+) hippocampal CA1 interneurons.**

| Experimental Data (post-mTBI vs. control) | | | | | Firing Rate Factor | Spike Width Factor | Spike Amplitude Factor |
|---|---|---|---|---|---|---|---|
| | | | | | 0.41 | 0.71 | 0.46 |
| **NEURON Modeling Conditions hippocampal PV+ basket cell (Bezaire et al., 2016)** | | | | | **Firing Rate Factor** | **Spike Width Factor** | **Spike Amplitude Factor** |
| | $th_{inf} = -59$ | | | | 0.40 | 0.72 | 0.73 |
| | $th_{inf} = -60$ | | | | 0.26 | 0.71 | 0.82 |
| | $th_{inf} = -59.5$ | $e_{Na} = 80$ | | | 0.39 | 0.70 | 0.97 |
| | $th_{inf} = -57$ | $e_{Na} = 80$ | $sh = 15.5$ | $g_{max}KD_{fast} = 0.026$ | 0.42 | 0.73 | 1.08 |
| | $th_{inf} = -57$ | | $sh = 15.5$ | $g_{max}KD_{fast} = 0.026$ | 0.39 | 0.77 | 0.93 |
| | $th_{inf} = -58$ | $e_{Na} = 70$ | | $g_{max}KD_{fast} = 0.026$ | 0.48 | 0.68 | 0.92 |
| | $th_{inf} = -59.3$ | $e_{Na} = 80$ | | | 0.42 | 0.71 | 0.98 |
| dendrites 1,2 | $th_{inf} = -59.3$ | | | | 0.42 | 0.74 | 0.88 |
| dendrites 0,1,2,3 | $th_{inf} = -59.3$ | $e_{Na} = 65$ | | | 0.45 | 0.74 | 0.90 |
| dendrites 0,1 | | $e_{Na} = 65$ | | | 0.45 | 0.74 | 0.91 |
| dendrites 2,3 | $th_{inf} = -59.3$ | $e_{Na} = 70$ | | | | | |
| dendrites 0,5,10,13 | $th_{inf} = -59.3$ | $e_{Na} = 60$ | | | 0.42 | 0.73 | 0.91 |
| dendrites 0,5,10,13 | $th_{inf} = -59.5$ | $e_{Na} = 65$ | | | 0.42 | 0.71 | 0.95 |

Parameters of the PV+ basket cell[45] were altered to simulate experimental data in post-mTBI vs. control groups of animals based on changes observed in CA1 interneurons at 7 days following mTBI. The best fit of the model to the experimental data was achieved by a change of a single parameter ($h_{inf} = -59$).

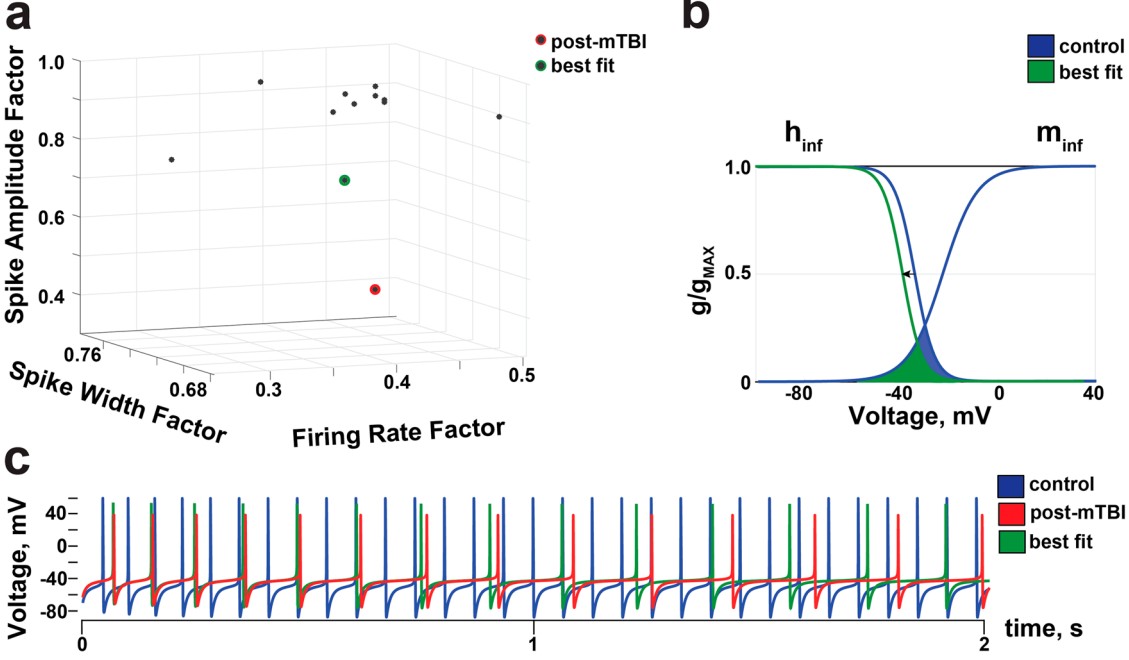

**Fig. 5 Model of basket cells reveals potential dysfunction in sodium channels. a** NEURON environment was used to model parvalbumin-positive (PV+) hippocampal CA1 interneurons[45]. The original parameters were altered in both the soma and dendrites to match experimental data from post-mTBI animals (red dot). The best fit (green dot) is achieved by altering a single parameter in the model ($th_{inf}$), and it corresponds to a 60% reduction in the interneuronal firing rate (vs. 59% in the experimental data), a 28% in the spike width (vs. 29% in the experimental data), and a 27% reduction in the spike amplitude (vs. 54% in the experimental data). See Table 1 for details. **b** Voltage dependence of activation ($m_{inf}$) and steady-state inactivation ($h_{inf}$) of sodium currents generated in the model PV+ CA1 interneuron ($g/g_{MAX}$ = normalized conductance). A best fit to the experimental data was produced by a negative shift of the inactivation ($h_{inf}$) from $-35$ mV (blue trace) to $-44.5$ mV (green trace) and is shown with an arrow. Activation ($m_{inf}$) was not altered when using the best fit parameters. The window current of voltage-gated sodium channels (area under the curves) was decreased by ~33% (green). **c** Changes in firing rate and spike amplitude shown for control (blue trace), post-mTBI (red trace), and best fit parameters (green trace).

concentration in axons[54]. Furthermore, axonal stretch injury induced structural changes in the alpha-subunit of voltage-gated sodium channels and led to changes in sodium channel inactivation, followed by a long-term increase in the intracellular $Ca^{2+}$ and subsequent proteolysis of the sodium channel alpha subunit[53]. Most recently, widespread and progressive loss of nodal axonal sodium channels have been reported in our pig model[55]. These results continue to implicate voltage-gated $Na^+$ channel dysfunction as a consequence of TBI, and our translational model of mTBI further suggests that this dysfunction may

preferentially affect interneurons in the CA1 region of the hippocampus.

The changes in oscillatory power and entrainment of CA1 neurons that we observed at 7 days following concussion may also suggest a potential shift in the recruitment/activation of CA1 neuronal subpopulations[56,57]. PV+ CA1 interneurons (specifically basket cells) are directly involved in feedforward inhibition and have been proposed to stabilize the frequency of local gamma oscillations and maintain a sparsity of neuronal ensembles in the presence of changes in the excitatory drive[33,39,40]. This observation may further support TBI associated disruptions in feedforward inhibition provided by PV+ interneurons in CA1. This disruption may also impair behavioral tasks requiring network sparsity (e.g., pattern separation), that have been previously reported to be disrupted in a rodent model of mTBI[58]. That concussion may lead to interneuronal changes underscores the importance of cognitive tasks that rely on spatiotemporal precision for clinical diagnosis. However, rigorously correlating these changes requires awake behaving electrophysiology and the development of behavioral tasks sensitive to this injury[24].

It is worth highlighting that the methods used to distinguish putative CA1 interneurons from putative CA1 pyramidal cells were developed based on activity in the rat hippocampus and are here adapted to the porcine hippocampus[18,28]. Electrophysiological characteristics that define spike shape and firing properties of pig hippocampal neurons may vary from those defining rodent CA1 neurons due to the differences in the size of neurons as well as variability in the voltage-gated channels responsible for the generation of action potentials[59]. Also, anesthesia has been known to affect neuronal firing rate as well as the pattern and strength of inputs arriving at the dendritic arbor of the CA1 pyramidal cells[60]. These caveats notwithstanding, the relative firing properties and spike shapes of pyramidal cells vs interneurons in CA1 follow the same patterns in swine as in rats with pyramidal cells typically firing in bursts (resulting in a smaller 1st moment of the autocorrelogram (Fig. 2a) and exhibiting lower firing rates and larger spike widths compared to interneurons which have more tonic firing patterns (resulting in a larger 1st moment of the autocorrelogram (Fig. 3a) and exhibit higher firing rates and more narrow spike shapes. Additionally, similarities between the laminar profile of oscillations (theta, gamma and ripple frequency bands) in the porcine and rodent CA1[25] support our use of these sorting techniques in this study.

Although we found no alterations in intrinsic properties, we demonstrated that CA1 pyramidal cells lose significant synchronization with local gamma oscillations at 7 days following concussion. Previously, we characterized the laminar profile of oscillations in the pig hippocampus[25]. Here, we observed a significant reduction in gamma oscillation (35–75 Hz) power detected locally to CA1 pyramidal cells post-concussion. The reduction in low gamma oscillations (35–55 Hz) could be due to a decrease in the fiber volleys of Schaffer collaterals (inputs from the CA3 layer) previously reported in this model[22]. Taking into account the fact that networks of interneurons in CA1 support the generation of local high gamma oscillations (55–75 Hz) and entrain CA1 pyramidal cells to these oscillations[33,61], these results support our findings above that CA1 interneurons are preferentially affected by inertial injury and that they have not fully recovered by 7 days following concussion. Due to the hypothesized role of theta-gamma coupling in memory and recall, loss of entrainment of CA1 pyramidal cells to gamma oscillations may underlie aspects of post-concussive syndrome[23]. However, oscillatory activity in this study is influenced by anesthesia, suggesting that these results need to be confirmed in the awake behaving animal.

In this study, we observed a prolong depolarization in a subset of injured animals (2 out of 9), followed by a loss of synaptic responses to stimulation. In epilepsy, paroxysmal depolarization shifts and spreading depolarization have been observed in the hippocampus and cortex using animal models as well as in human patients[34]. In TBI, spontaneously generated paroxysmal depolarizations have been only detected in organotypic hippocampal cultures following long-term activity deprivation, potentially due to a homeostatic plasticity mechanism[62]. Therefore, the prolonged depolarizing shift observed using our model of concussion is the first such epileptogenic event described in the in vivo preparation to the best of our knowledge. A long-lasting depolarization shift and accompanying repetitive action potentials following paired-pulse stimulation of Schaffer collaterals could be due to changes in fiber volleys, given the differential response to varying stimulation intensities in slice previously reported between control and injured animals using this model[22]. Additionally, we also observed recurring paroxysmal oscillatory events in the form of arcuate spikes occurring in a synchronized manner across all hippocampal layers in all post-mTBI but not control animals. Similar epileptogenic spikes have been previously reported after experimental TBI in rodent models and are thought to reflect repeated population spikes associated with hypersynchronous multi-unit discharges[63]. Both findings could be substantially influenced by the loss of activity of hippocampal interneurons. Therefore, there are potentially multiple pathological mechanisms involved in the generation of the prolonged depolarizations since we are observing different inputs to the CA1 dendritic arbor simultaneously with these laminar probes.

Concussion has been previously identified as a risk factor for the development of epilepsy[64]. However, mechanisms underlying the development of post-traumatic epileptogenesis following concussion have not yet been identified. Here, we reported a significant reduction in the power of high frequency oscillations (80–300 Hz) detected locally to interneurons (but not pyramidal) CA1 cells, which may be due to the observed reduction in the firing rate of CA1 interneurons or desynchronization of the CA1 network. A reduction in the number of hippocampal interneurons is a prominent pathology associated with the development of unprovoked seizures both in experimental models and in human temporal lobe epilepsy[65]. While interneuronal cell loss in our porcine inertial injury model has not been histologically assessed, the number of electrophysiologically active cells recorded in our model was not significantly different between the control and injured animals. Nevertheless, changes in the electrophysiological properties of interneurons may affect circuit level hippocampal interactions and increase the risk for hyperexcitability, and therefore potentially epileptogenesis.

Hippocampal CA1 hyperexcitability and epileptogenesis caused by changes in sodium channels expressed in inhibitory PV+ interneurons have been previously reported[66]. While histological and electrophysiological studies have previously identified PV+ interneurons to be selectively altered in experimental models of rodent TBI[67–70], we could not identify a specific sodium channel responsible for the observed changes in CA1 interneurons following concussion. Using NEURON modeling of a CA1 PV+ interneuron (specifically a basket cell), we observed a reduction in the sodium channel window current which in turn may also affect resurgent and/or persistent currents generated by voltage-gated sodium channels[71]. Interestingly, changes in the window current of voltage-gated sodium channels have been previously reported in epilepsy[72,73]. Potential mechanisms for the negative shift of sodium channel inactivation include altered properties of sodium channel auxiliary subunits (beta[74] and gamma subunits) as well as the involvement of slow inactivation caused by prolonged depolarization[75], both of which were

proposed to play a crucial role in sodium channel associated epilepsies[72]. In addition, multifocal disruption of the blood brain barrier (BBB) has been demonstrated in this model up to at least 72 h post-mTBI as evidenced by extravasation of serum proteins, fibrinogen, and immunoglobulin G, and could potentially contribute to observed electrophysiological changes[76].

There are several limitations to our electrophysiological approach, including the anesthetized preparation. The predominant effects of isoflurane are known to be on GABA, glycine, and NMDA receptors, and we cannot rule out differential effects on interneurons vs. pyramidal cells[77,78]. However, we have no evidence to suggest that anesthesia would differentially affect intrinsic properties or synaptic release probabilities of injured interneurons vs. pyramidal neurons post-injury[79,80]. Differences in metabolic states may also contribute to the observed electrophysiological changes, along with potential injury-induced vascular remodeling or changes in neurovascular coupling[81]. We selectively focused on hippocampal area CA1 due to our prior results demonstrating profound changes in this region in a hippocampal slice preparation with this model[22], as well as the known role of the hippocampus in various aspects of cognition and its selective vulnerability in TBI and post-traumatic epilepsy. Verification of these data in awake, chronically implanted animals and expansion to other limbic areas involved in these processes is important future work[82].

In summary, using a translational, large animal model of concussion we have demonstrated that a single inertial injury alters hippocampal circuitry at 7 days post-injury by affecting the electrophysiological properties of CA1 neurons. In particular, the physiological properties of interneurons in CA1 are substantially altered following concussion. These interneuronal alterations may be due to a change in the inactivation of voltage-gated sodium channels expressed in the somata of PV$^+$ basket cells and may underlie the exhibited decrease in local gamma power and the loss of entrainment of pyramidal cells to local gamma oscillations. We also observed hippocampal hyperexcitability in the local field potentials, predominantly following stimulation. These changes may underlie aspects of cognitive dysfunction following concussion, as well as increase the risk for post-traumatic epilepsy[16]. Future studies are needed to investigate if the changes in sodium channel function predicted by our model are observed experimentally. Further assessing these changes over time and determining the differential involvement of specific interneuron populations may reveal therapeutic targets for improving concussion induced memory dysfunction and/or epileptogenesis. Restoring GABAergic interneuronal function before hyperexcitability proceeds to epileptogenesis may improve cognitive outcomes and decrease seizure frequency, as has been previously demonstrated in a mouse model of post-traumatic epilepsy[83]. While it is unknown whether the observed changes in interneuronal function persist at time points greater than one-week post-injury, pharmacological interventions centered on restoring sodium channel inactivation may be a potential target for improving cognitive function following concussion.

## Methods

**Animal care.** Yucatan miniature castrated male pigs ($n = 17$) were purchased from Sinclair and underwent the current studies at the age of 6 months and a mean weight of $29.8 \pm 1.3$ kg (mean ± SEM). The animals were fed standard pig chow and water ad libitum. All animals were pair housed when possible and were always in a shared room with other pigs in a research facility certified by the Association for Assessment and Accreditation of Laboratory Animal Care International (AAALAC facility). All animal experiments were approved by and conducted according

to the ethical guidelines set by the Institutional Animal Care and Use Committee of the University of Pennsylvania and adhered to the guidelines set forth in the NIH Public Health Service Policy on Humane Care and Use of Laboratory Animals (2015).

**Surgical procedures.** Prior to the procedures, animals were fasted for 16 h with water remaining ad libitum. After induction with 20 mg/kg of ketamine (Hospira, 0409-2051-05) and 0.5 mg/kg of midazolam (Hospira, 0409–2596-05), anesthesia was provided with using 2–2.5% isoflurane (Piramal, 66794-013-25) via a snout mask and glycopyrrolate was given subcutaneously to curb secretions (0.01 mg/kg; West-Ward Pharmaceutical Corp., 0143-9682-25). The animals were intubated with a size 6.0 mm endotracheal tube and anesthesia was maintained with 2–2.5% isoflurane per 2 liters $O_2$. Heart rate, respiratory rate, arterial oxygen saturation, end tidal $CO_2$, blood pressure and rectal temperature were continuously monitored, while pain response to pinch was periodically assessed. All these measures were used to maintain an adequate level of anesthesia. A forced air warming system was used to maintain normothermia.

**Closed-head rotational injury using the HYGE pneumatic actuator.** An injury was induced according to the previously described protocols[84]. Briefly, the injury was performed under isoflurane anesthesia, while physiological parameters of the animals were being closely monitored. Closed-head rotational acceleration was performed using the HYGE pneumatic actuator, a device capable of producing non-impact head rotation (up to 110 degrees in less than 20 ms) with a controlled relationship between maximum rotational acceleration and injury severity. Angular velocity was recorded using a magneto-hydrodynamic sensor (Applied Technology Associates, Albuquerque, NM) connected to a National Instruments DAQ, controlled by Lab-VIEW. The animal's head was secured to a padded bite plate under anesthesia and attached to the HYGE device. A single rapid head rotation was performed within a range of rotational velocities (168–269 rad/s), with the mean angular peak velocity being $226 \pm 13$ rad/s (mean ± SEM, $n = 9$). For a subset of analyses, the injured animals were divided into two subgroups, those injured at ~190 rad/s ($188 \pm 11$ rad/s, mean ± SEM, $n = 4$) versus those injured at ~260 rad/s ($256 \pm 7$ rad/s, mean ± SEM, $n = 5$). The injury model has been extensively characterized in previously published studies[76,85,86]. Control animals (total $n = 8$) included naïve animals ($n = 4$). Another subset ($n = 3$) received an additional craniectomy anterior to the recording site as control for another study (ML = 9, AP = +13). A single sham animal underwent all injury procedures 7 days prior to the recording, including attachment to the HYGE device without firing it, and was kept under anesthesia for the same amount of time as injured animals.

**Laminar multichannel in vivo neurophysiological recordings in porcine hippocampus.** Electrode implantation was performed in all control and post-mTBI animals ($n = 17$ total) according to the previously described protocols[24,25]. Briefly, pigs undergoing electrophysiological recordings (control: $n = 8$ animals; post-TBI: $n = 9$ animals) were placed in a stereotaxic frame prior to electrode placement. Large animal stereotaxis combined with single-unit electrophysiological mapping via a tungsten monopolar electrode (Cat# UEWSEGSEBNNM, FHC) was used to precisely place a 32-channel laminar silicon probe into the CA1 region of the dorsal hippocampus. For laminar multichannel recordings, either a NN32/LIN linear 32-channel silicon probe (100 µm or 300 µm spacing between individual channels, Cat# P1x32-70 mm-100/300-314r-HP32, NeuroNexus) or a NN32/TET custom-

design 32-channel silicon probe (275 µm spacing between individual tetrodes, Cat# V1x32-15 mm-tet-lin-177, NeuroNexus) was used for electrophysiological recordings (see ref. [24] for more details on multichannel silicon probes used in this study). NN32/LIN probes were used in 9 animals (control = 5 vs. post-mTBI = 4), and NN32/TET probes were used in 8 animals (control = 3 vs. post-mTBI = 5). A skull screw was placed over the contralateral cortex as a reference signal.

Laminar oscillatory field potentials and single-unit activity were recorded concurrently in the dorsal hippocampus of pigs under isoflurane anesthesia (2–2.5%). Anesthesia Neurophysiological signals were amplified and acquired continuously at 32 kHz on a 64-channel Digital Lynx 4SX acquisition system with Cheetah recording and acquisition software (Neuralynx, Inc.). The final location of the silicon probes was confirmed electrophysiologically as described previously[25]. Briefly, the pyramidal CA1 layer was identified by calculating the root mean squared power in the 600–6000 Hz frequency band, a proxy for the spiking activity of neurons.

**Electrical stimulation**. In a subset of animals (control = 3, post-mTBI = 7), electrical stimulation was performed in the Schaffer Collaterals using custom-designed concentric bipolar electrodes (Cat# CBBPC75(AU1), FHC) to investigate CA1 responses to the inputs coming in from the hippocampal CA3 layer. A-M Systems Model 3800 stimulator with MultiStim software installed was used to generate electrical pulses, while A-M Systems Model 3820 Stimulus Isolation Unit was used to deliver the current. Input-output stimulation in the range of 100–1000 µA was performed prior to the experimental stimulation for each animal, and the final stimulation amplitude (range 200–500 µA) was selected at the half maximum response to stimulation[25]. Stimulation amplitude was not significantly different between the control and post-mTBI groups of animals (control: $n_{animals}$ = 4, Stimulation Amplitude = 325 ± 66 µA vs. post-mTBI: $n_{animals}$ = 7, Stimulation Amplitude = 386 ± 54 µA, $p$ = 0.5061).

At the end of electrode insertion procedure, all pigs underwent transcardial perfusion under anesthesia using 0.9% heparinized saline followed by 10% neutral buffered formalin (NBF). Brains were extracted and inspected for gross pathology. Minimal subdural and/or subarachnoid blood regionally associated with electrode insertion was appreciated in all animals. In three animals (control = 1, post-mTBI = 2), minimal subarachnoid blood was also observed ventral to the brainstem. No gross evidence of brain swelling was identified, consistent with prior descriptions of the model.

**Single unit analyses**. Off-line spike detection and sorting was performed on the wideband signals using the Klusta software packages (http://klusta-team.github.io/klustakwik/), and clusters were manually refined with KlustaViewa software (https://github.com/klusta-team/klustaviewa)[87]. The number of single units per animal recorded with either type of multichannel silicon probes (NN32/TET vs. NN32/LIN) was not significantly different (Supplementary Fig. 1c), with total number of single units recorded in control ($n_{cells}$ = 125) and post-TBI ($n_{cells}$ = 93) groups used for comparison between the groups. The number of recorded cells per animal was not different between the groups (control = 16 ± 3 cells vs. post-mTBI = 10 ± 3 cells, $p$ = 0.2397). There were no significant differences detected between a number of single units recorded with two types of electrodes (NN32/TET = 12 ± 3 cells vs. NN32/LIN = 14 ± 3 cells per animal, $p$ = 0.6452) or between a number of single units recorded with each type of electrode for control vs. post-mTBI groups of animals (NN32/TET: control = 16 ± 6 vs. post-mTBI = 9 ± 3 cells

per animal, $p$ = 0.2608; and NN32/LIN: control = 15 ± 5 vs. post-mTBI = 12 ± 6 cells per animal, $p$ = 0.6937). Resulting single-unit clusters were then imported into Matlab (version R2023a) for visualization and further analyses using custom and built-in routines (https://www.mathworks.com/products/matlab.html). In order to minimize waveform shape distortion for visualization, the wideband signal was high pass filtered using a wavelet multi-level decomposition and reconstruction filter (level 6, Daubechies 4 wavelet)[88]. Waveforms were then extracted from this filtered signal and averaged for display. The freely available Matlab packages FMAToolbox (http://fmatoolbox.sourceforge.net[48]) and Chronux (http://chronux.org[48,89]) were used for analysis of neuronal bursting properties.

Firing rate (in Hz) was calculated as the average number of spikes per second. All firing rates reported here are of spontaneously active neurons recorded under anesthesia. For each unit, the autocorrelogram was calculated by binning spike times into 1 ms bins and computing the correlation of the binned spike times at lags from 1 ms to 50 ms. The counts in each bin were divided by the total number of counts to create a probability distribution that is used for plotting. The first moment of the autocorrelogram (in ms) was defined as the mean value of the autocorrelogram in time. Spike amplitude (in µV) was calculated as a distance from baseline to the highest peak. Spike width (in ms) was calculated at the half point of spike amplitude.

**Putative neuron subtype classification**. Hippocampal CA1 cells were classified into groups of putative pyramidal cells and inhibitory interneurons[27] following previously described protocols[18,28]. Briefly, the following criteria were used: 1) anatomical location: only single units from the top portion of the multichannel silicon probe were selected, which anatomically corresponded to the pyramidal CA1 cell layer; 2) firing rate threshold: putative pyramidal CA1 cells had firing rate of 7 Hz or below, while CA1 interneurons had firing rate above 7 Hz (under anesthesia); and 3) spike waveform and autocorrelogram: first moment of autocorrelogram and symmetry of spike waveforms were used to manually place cells into putative CA1 pyramidal cells vs. CA1 interneurons category similarly to previously published criteria for rodent studies[18,28]. We also performed the automated clustering of pyramidal cells and interneurons using Matlab K-means clustering function (with k = 2 clusters) as previously described in the rodent literature[28] and compared the results of automated vs. manual clustering (Supplementary Fig. 1d, e). While output from the automated clustering was highly similar to our manual clustering results, it moved some obvious neuronal sub-types into incorrect categories, and manually identified clusters were used for the final analysis.

**Local field potentials analyses**. Acquired wideband local field potentials recorded with the 32-channel silicon probe were downsampled to 2 kHz for further analysis. Signals were imported into Matlab software, version R2023a and processed using a combination of custom and modified routines from the freely available Matlab packages FMAToolbox (http://fmatoolbox.sourceforge.net[48]), Chronux (http://chronux.org[48,89]), EEGLAB (http://sccn.ucsd.edu/eeglab/index.html[90]), and CSDplotter, version 0.1.1[91]. Entrainment strength was determined by computing the modulation vector length (MVL) each single unit with oscillations ranging from 1 to 300 Hz on the channel that contained the maximal spike amplitude. Entrainment strength for all single units was then averaged and compared between the control and post-mTBI groups. Power spectrum density analysis of hippocampal oscillations was performed for all channels with detectable single units as previously described[25]. If more than one

single unit was recorded on the same channel, the duplicate power spectrum was excluded from the analysis. The calculated power spectrum density was averaged across all animals then z-score normalized for comparative analysis between the control and post-mTBI groups of animals.

**NEURON modeling**. The NEURON simulation environment was used to model the concussion-associated electrophysiological changes observed in CA1 interneurons[43,44]. Specifically, a model of parvalbumin-positive (PV+) basket cells, fast-spiking CA1 interneurons that synapse on the somata and proximal dendrites of CA1 pyramidal cells, was used as described previously[45,92]. The individual parameters of the model (activation/inactivation and the maximum conductance of voltage gated channels, ionic driving force, action potential amplitude, number of dendrites per neuron, etc.) were altered either alone or in a combination with other parameters in a blinded manner to fit the experimental data (summarized in Table 1). The action potentials of the model-generated interneurons were generated and then compared to the experimentally observed values of the firing rate, spike amplitude and spike width following concussion. The parameter(s) most likely to be affected by the injury were identified based on the best fit between experimentally observed data and the NEURON model generated values.

**Statistics and reproducibility**. Electrophysiological data were collected in 17 animals total (control = 8, post-mTBI = 9). Total number of single units in control ($n_{cells}$ = 125) and post-TBI ($n_{cells}$ = 93) groups were used for comparison. The data were analyzed using Matlab (version R2023a, custom and built-in routines) as well as Graphpad Prism (version 10) software. Non-parametric Mann-Whitney test was used for comparative analyses of neuronal properties (firing rate, spike width, and spike amplitude) between the control and post-mTBI groups of animals. Data were displayed as mean ± SEM. For a subset of analyses of neuronal properties, post-mTBI animals were divided into two subgroups, injured at ~190 rad/sec (n = 4) vs. at ~260 rad/sec (n = 5). Ordinary one-way ANOVA was used for comparative analysis between the control, ~190 rad/sec, and ~260 rad/sec groups of animals. For power analyses, two sample t-test was used for comparison, and data were presented as mean ± STD. For entrainment analyses, non-parametric Mann-Whitney test was used for comparison, and data were presented as mean ± SEM. The threshold for significance was used with $p$-value of less than 0.05.

**Reporting summary**. Further information on research design is available in the Nature Portfolio Reporting Summary linked to this article.

## Data availability

The electrophysiological data sets generated and/or analyzed during the current study are available from the corresponding author on reasonable request. The numerical source data for Figs. 1–3 are publicly available on FigShare[93].

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

## Acknowledgements
DoD W81XWH-22-1-0287 (A.V.U.), NIH NINDS T32-NS043126, VA RR&D I01-RX001097 (D.K.C., J.A.W.), DoD W81XWH-20-1-0838 (V.E.J.), DoD W81XWH-20-1-0901 (J.A.W.), VA RR&D RX003498 (J.A.W.), VA RR&D RX002705 (J.A.W.), NIH R01 NS101108 (J.A.W.).

## Author contributions
A.V.U., D.K.C., V.E.J., and J.A.W. designed and performed experiments, analyzed data, and wrote the paper; C.C., C.D.A., O.E.F., and P.F.K. developed analytical tools and analyzed data; N.M. designed and implemented in silico model; C.D.A., J.A., M.R.G., and P.F.K. gave technical support and conceptual advice; A.V.U., C.D.A., C.C., N.M., M.R.G., O.E.F., J.A., P.F.K., V.E.J., D.K.C., J.A.W. edited the manuscript; D.K.C., V.E.J., and J.A.W. raised grant funds for the study; J.A.W. supervised the study.

## Competing interests
The authors declare no competing interests.

## Additional information

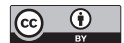

