## [Peer Review File · Communications Biology]

Reviewers' comments:

Reviewer #1 (Remarks to the Author):

In this manuscript titled "Hippocampal Interneuronal Dysfunction and Hyperexcitability in a Porcine Model of Concussion", the authors detected electrophysiological changes of hippocampal network at 7 days post-injury in anesthetized minipigs. They found that firing rate and burst occurrence of CA1 interneurons were significantly decreased post-mTBI, and were unchanged in CA1 pyramidal cells. Furthermore, they found that pyramidal CA1 neurons in TBI animals were significantly less entrained to hippocampal gamma (40 - 80 Hz) oscillations. They concluded that a single concussion can lead to significant neuronal and circuit level changes in the hippocampus, which may be important contributors to cognitive dysfunction following mTBI. This study provides some interesting findings. I have the following suggestions before the manuscript can be suitable for publication.

1. Whether this single injury protocol can induce long-term cognitive dysfunction is not clear per their previous reports. What is the rationale to detect changes at 7 days after injury to predict the long-term cognitive dysfunction, which might happen months in preclinical models after injury? What if it is just a transient response of the body in the early phase of injury and have nothing to do with the long-term cognitive dysfunction?
2. If data in Figure 1 include all those data in Figure 2 on pyramidal neurons and 3 on interneurons, what is the purpose to show the Figure 1.
3. Those data showing in Figure 1, 2 and 3 are confusing. What does the firing rates in Figure 1,2,3 B mean, evoked or spontaneous? What does the burst per minute mean, is it still firing rate? Please show sample traces for each recording that readers can understand what is recorded and analyzed.
4. Is the spike in Figure 1-3B evoked field potential or spontaneous firing? If evoked field potential, stimulation intensity needs to be provided.
5. The way to distinguish pyramidal neurons and interneurons is from studies in rats. Please provide citation or confirm hippocampal neurons from rat and pig share the similar electrophysiological properties.
6. In Figure 4 on evoked field potentials, they found that "a sustained depolarization over 1 sec long was observed in a subset (n=2 out of 5) of animals injured at ~ 260 rad/sec". Please clarify on how the depolarization was measured or observed, as this is extracellular field potential recording. What comparison was used to get the conclusion that those pyramidal neurons are more excitable than pig with sham treatment?
7. The discussion is way too long and out of focus.

Reviewer #2 (Remarks to the Author):

Manuscript by Ulyanova, Cottone, Adam, Maheshwari, Grovola, Fruchet, Alamar, Koch, Johnson, Cullen, and Wolf entitled "Hippocampal Interneuronal Dysfunction and Hyperexcitability in a Porcine Model of Concussion" present interesting evidence for the abnormal laminar oscillatory field potentials and single unit activity in the intact hippocampal network at 1 week post-concussion in anesthetized minipigs. The found evidence are somewhat consistent with hyperexcitability of pyramidal cells and interneurons in the area CA1. Computational simulations suggest that these changes in interneuronal firing and waveforms may be due to alterations in gating mechanism of voltage-gated sodium channels. This work is an interesting extension of the previously reported findings from ex vivo slice recording. In the new study hippocampal networks were assessed using multichannel probes designed for recording from deep brain structures, oscillatory field potentials and single unit activity were analyzed.

There are few major concerns with regard to the manuscript as its presentation to the readers:

- 1) The *in vivo* finding should provide expanded/broader evidence for remodeling in hippocampal network, this should include both electrical activity and putative contribution from vascular supplies. There are no data showing that vascular/oxygen supply to hippocampus is not altered by injury.
- 2) Does anesthesia and its level impact the electrophysiological activity in the similar manner in control and injured animals?
- 3) What are behavioral evidence for hippocampal impairments?
- 4) Why there are no anatomical/pathological assessment of injury variability and sodium channel localization/re-localization post-injury? Can channel translocation or reassembly be considered as an alternative explanation? Were any specific post-translational modifications noted?

Specific comments:

a) Animal were fasted for 16 hours previously to recording. There are no data showing level of glucose in both group of animals and how this metabolic impact affects the physiological response, what is then impact of anesthesia and how this affects recordings?

b) In the Method Section the authors state: "The final location of the silicon probes was confirmed electrophysiologically as described previously (Ulyanova, Koch et al. 2018)."

The functional methodology is described but without paralleled anatomical/stereotaxic description this consideration appeared too general and provide readers of TBI literature very limited knowledge of location with respect to injury and animal brain anatomy.

The comments C-F derive from the lack of description of methodology in order to provide in depth understanding of results and their impact for a reader who is not computational physiologist. The authors refer to their previous methodological papers but they do not try simplify/describe their rational so wide spectrum of TBI oriented audience can understand analytical details in terms of anatomy and neurophysiology.

c) In the Figure 1: the authors present coronal section but there is no clarity where are these sections from. 3Dim description of placement of stimulation and recording electrodes would be required so readers have a better knowledge of recording localization. How did authors control for similarity of electrode placement and how injury pattern was included in these considerations? What statistical analysis have been used and how cell numbers have been approximated and compared between injury models? There is always variability in the injury pattern how the reader can be assured that this variability does not impact the final conclusions.

d) In the Figure 2: the authors compared power spectra but there are no details how and why these analysis were carried out. Were input-output curves similar? Were response variabilities assessed? Did changes in the entrainment impact the amount of information that spike rates transmitted over gamma-cycle time windows? How much output variability was unrelated to input? Why only functional assessment of the cells is presented? Can morphology of cells be investigated following the recording? What statistical analysis have been used and how cell numbers was approximated and compared between injury models?

e) For Figure 3, similar comments as for Figure 2.

f) In the Figure 4 some traces are presented but it is not clear if these traces are from different or same animals? This figure as it is presented seems to be descriptive and speculative. More detailed justifications would be required to match these outcomes and what is quantified in Figures 1-3. At minimum, the origin (animal and injury) of all traces have to be specified.

g) In the Figure 5 the authors apply computational model to the basket cells? Are there any experimental evidence to support that this interneuron-type has been selected? Any anatomical evidence? The authors state: "Under these modeling conditions, the window current of voltage-gated sodium channels, defined as an overlap between the activation (*minf*) and inactivation (*hinf*) curves, is decreased by ~30% following inertial injury." This is potentially important conclusion but no direct data are offered to support this conclusion and the potential underlying mechanism.

Reviewer #3 (Remarks to the Author):

The paper by Ulyanova et al. examined how hippocampal multiunit activity, especially in interneurons, is altered in a porcine model of concussion. Their experiments were nicely performed. In addition, they created a reliable mathematical model to speculate a molecular mechanism. As concussion is a crucial brain injury, I believe that this report provides crucial evidence for broad audience by showing that this pathological change may underlie memory dysfunction. However, I have several concerns that need to be clarified as below.

(1) Introduction: I could not well understand the originality of this study compared with rodents. Why they needed to utilize minipigs as an animal model? While they briefly described unknown questions in line 83-85, how this issue is related to the use of minipigs? Please add more information for the advantages of this animal species.

(2) My crucial concern is their analysis after the classification of neuron types. In the Methods, they described a threshold for classification is "7 Hz". This value itself is no problem. But they again compared firing rates of each cell type (Fig. 2 and 3) after this classification. This may be a problem because they applied statistics on a parameter after the classification based on that parameter. As shown in Figure 1, overall firing rates of all neurons are reduced by concussion. This means that some interneurons with reduced firing rates (below 7 Hz) may be misclassified as excitatory cells. If this problem happens, their results in Figure 2 (no changes in excitatory cells with a firing rate of <7 Hz) may represent just an artifact of this classification. Nonetheless, I think their results in Figure 3 are still meaningful because it demonstrates reduced firing in cell groups with a higher firing rate of >7 Hz (possibly all interneurons).

As it is not easy to solve this statistical problem because the data were obtained from independent animal groups, at least, I suggest that they present all the group-comparison datasets by distributions of individual datasets (not bar graphs as they presented).

In addition, as this classification is a central issue of this study, please fully described all criteria and parameters used (not just "previously published criteria" on Line 652-663).

(3) Figure 2C, D, Figure 3C, D: I think LFP power analysis on frequency bands less than 100 Hz is meaningful. However, I do not understand the meaning of "overall LFP power" on ripple-band (150-250 Hz). Generally, we directly detect transient "ripple events" based on RMS or Hilbert transformation, which will be more useful to discuss how memory consolidation and retrieval mechanisms are altered. I suggest that they employ similar ripple analyses to previous studies (e.g. the frequency of ripple events, amplitude of ripple events, and ripple-triggered spike rates).

(4) Figure 4: These graphs and related sentences have no quantification. Please quantify these data and describe their claims based on statistical results. For example, fEPSP amplitude (Fig. 4A), duration or area of sustained depolarization (Fig. 4B), and the number of unit signals (Fig. 4C) could be quantified. By the way, why they applied paired pulse stimulation? Generally, this stimulation is utilized to analyze presynaptic release. Please describe changes in paired pulse ratio to discuss synaptic changes.

(5) Line 614-617: They used two types of silicon probes (NN32/LIN and NN32/TET). To my experience, NN32/LIN is really useful to capture multiunit signals? (as recording points are linearly located with a large interval) Please clarify how many cells or how many LFP signals were captured by these two types of how many probes in the Methods or Results part.

Title: "Hippocampal Interneuronal Dysfunction and Hyperexcitability in a Porcine Model of Concussion"

Manuscript: COMMSBIO-22-0908A

Response to Reviews:

We are grateful for the thorough reviews that were provided to us on our manuscript, as well as the detailed suggestions for improvement. We believe we have been able to address the reviewers' suggestions in this revision, and that the manuscript is significantly strengthened as a result. Below is our point-by-point response to the reviewers' comments:

Reviewer 1 Comments:

1. *Whether this single injury protocol can induce long-term cognitive dysfunction is not clear per their previous reports. What is the rationale to detect changes at 7 days after injury to predict long-term cognitive dysfunction, which might happen months in preclinical models after injury? What if it is just a transient response of the body in the early phase of injury and has nothing to do with long-term cognitive dysfunction?*

This is indeed a valid question. We have been working for the last 5 years to develop cognitive tasks that are sensitive enough to detect cognitive changes post mild and moderate traumatic brain injuries (TBIs) in our pig models. Existing traditional behavioral tasks (hole-board, etc.) are far too easy for pigs to detect a difference in even a moderate TBI, necessitating more complex tasks. We have therefore developed complex touchscreen-based tasks including a Conditional Association Task (CAT), which requires pigs to not only choose appropriate responses, but to do the reverse of the prior response when a conditional cue is presented. Our pigs can perform this task at 70% correct prior to injury, and we have demonstrated preliminary deficits in this task in a controlled cortical impact (CCI) injury model recently, which features both focal and diffuse injury components (unpublished). We fully intend to characterize deficits in the rotational, mild TBI model as well out to 6 months post injury, but this is beyond the scope of the current manuscript. However, we have demonstrated chronic neuropathological changes out to 1 year in this injury model, particularly in the hippocampal region (Grovola et al. 2021). We agree it is an important question to validate whether chronic changes in hippocampal neuronal dysfunction A) correlates with behavioral changes out to chronic timepoints and B) is causal to any detected deficits, and we are working to collect that data.

The logic in *discussing* chronic cognitive dysfunction post-injury is as follows (we obviously present no supportive data yet): A) concussion is known to lead to acute and chronic cognitive dysfunction in a subset of mTBI patients, B) this is the best model of inertial injury we have, C) there are acute changes (7 days post) in the hippocampus that may underlie cognitive dysfunction, and D) these changes may persist. We fully agree that we present no data to support the correlation between these acute changes and cognitive changes and have toned down the discussion of this in the manuscript and appropriately labeled it as speculation (page 15, lines 364-367).

2. *If data in Figure 1 include all those data in Figure 2 on pyramidal neurons and 3 on interneurons, what is the purpose of showing the Figure 1?*

The main purpose of Figure 1 is to show how the data were collected in this study and to present our initial observations of how injury affects *all* pig hippocampal neurons (page 4-5, lines 112-124). The main result from these experiments was changes in the electrophysiological properties of hippocampal

neurons at 7 days following injury. We believe that it is important to show this in Figure 1 prior to attempting to separate pyramidal vs. interneurons using various methodologies (electrophysiological features and location on the electrode).

In addition, in response to another Reviewer's request, we updated Figure 1A by adding an image of the whole brain and a coronal section to make this figure clearer (page 5, line 125). We also added Figure 1B (page 5) to show examples of electrophysiological signals recorded in this dataset (local field potentials and single units). We moved the graphs to Figure 1C and modified them to show individual data points on the graphs instead of simple columns (as per Reviewer's suggestion). The Methods section is now updated to include detailed description of the electrophysiological parameters calculated in the dataset: firing rate, spike width, spike amplitude, and first moment of the autocorrelogram (pages 25, lines 660-672).

3. Those data showing in Figures 1, 2 and 3 are confusing. What does the firing rates in Figure 1,2,3 B mean, evoked or spontaneous? What does the burst per minute mean, is it still firing rate? Please show sample traces for each recording that readers can understand what is recorded and analyzed.

Thank you for pointing out that the data presentation on Figures 1-3 was confusing. In order to address this point, we have included a more detailed description in the Methods section of what is recorded and being analyzed (pages 23-24, lines 607-6624; page 26, lines 634-672). Specifically, single units are sorted out and identified from the baseline recordings, without any electrical stimulation applied in the pig hippocampus. All firing rates shown in Figure 1-3 are spontaneous activity of the neurons recorded under anesthesia. We updated the main text and Figures 1-3 to include a more detailed description of electrophysiological properties calculated properties (as described in Methods section, pages 25, lines 660-672). We have also updated Figure 1 A and B to include representative traces of local field potentials and spontaneous single units (page 5).

4. Is the spike in Figure 1-3B evoked field potential or spontaneous firing? If evoked field potential, stimulation intensity needs to be provided.

Thank you for pointing out this to us. All single unit data presented in Figures 1-3 are spontaneously recorded, no stimulation has been applied. We added this information to the main text, the corresponding Figure legends, and the Methods section. Only Figure 4 shows evoked field potentials in response to stimulation, and we included stimulation intensities to Figure 4 (and its legend) as per reviewer's suggestion (page 11, line 261).

5. The way to distinguish pyramidal neurons and interneurons is from studies in rats. Please provide citation or confirm hippocampal neurons from rat and pig share the similar electrophysiological properties.

This is an interesting point, and one that concerned us as well. Our laboratory was the first to begin to characterize pig hippocampal neurons (Ulyanova, 2018). Interestingly, the pig hippocampus appears to bridge the gap between rodent and primate hippocampus in both architecture (tight CA1 layer in rodents vs. dispersed in primates/pigs) and neurophysiology (theta at 7.7 Hz in rodents vs. 6 Hz in pigs). We are in the process of characterizing extensively the awake neuronal activity of hippocampal neurons in chronically implanted animals (also see (Ulyanova, 2019)). Pyramidal cell and interneuronal

activity from the pigs in these preparations follow similar patterns as in rodents and primates, but with slightly lower firing rates in pyramidal neurons. For example, interneurons tend to have higher firing rates and more narrow action potentials compared to pyramidal cells. Additionally, pyramidal cells tend to fire in bursts compared to more tonic firing of interneurons. Spike architecture is in the process of being extensively characterized, and it appears to be closer to primates than rodents. However, we do not expect that the methodology used in rodent studies (an algorithm where firing rate, width, and first moment of autocorrelogram are the features used for clustering) to vary from the rodent results, as it is not dependent on the absolute values of these measures but rather the overall patterns (see below for more detail explanation in our response to Reviewer #3).

6. *In Figure 4 on evoked field potentials, they found that “a sustained depolarization over 1 sec long was observed in a subset (n=2 out of 5) of animals injured at ~260 rad/sec”. Please clarify on how the depolarization was measured or observed, as this is extracellular field potential recording.*

To characterize an event as “depolarization”, we used standard convention for extracellular EEG recordings as described in the review article by Buzsaki et. al. (Nat Rev Neurosci. 2012 Jun; 13(6): 407–420). Specifically, extracellular potentials are generated by depolarization can be identified if local field potentials in the pyramidal CA1 layer have negative deflections over baseline (vs. hyperpolarization-induced positive deflection of local field potentials). We modified Figure 4 to include a baseline to visualize negative vs. positive deflections better (page 11). We also added a citation to this review article to the main text of the manuscript for reference (page 10, line 250).

7. *The discussion is way too long and out of focus.*

Thank you for your comment. As per your suggestion, we have edited the discussion section to focus it further on the main findings described in the paper (pages 14-21).

Reviewer #2 TBI, hippocampal networks (Remarks to the Author):

1) *The in vivo finding should provide expanded/broader evidence for remodeling in hippocampal network, this should include both electrical activity and putative contribution from vascular supplies. There are no data showing that vascular/oxygen supply to hippocampus is not altered by injury.*

This is an important point – we believe that the electrophysiological activity of these neurons is impacted by vascular changes as well, and not just sodium channel alterations. We have added a sentence to the discussion pointing out that we have previously characterized blood-barrier breakdown in this model at early timepoints (Johnson, V. E. et. al., *Acta Neuropath* 2018) and proposing other potential mechanisms for these changes (page 20, lines 510-514).

2) *Does anesthesia and its level impact the electrophysiological activity in the similar manner in control and injured animals?*

This is an intriguing question, but unfortunately one that is impossible to tease out. We have no reason to believe that anesthesia impacts activity differently in injured animals, as evidenced by the parameters that are unchanged between the control and injured preparations. We don't believe that a specific impact of the anesthesia on injured interneurons is likely. We are currently working on the awake behaving pig electrophysiology study, and we hope to be able to answer these questions in the future.

3) *What is behavioral evidence for hippocampal impairments?*

Please see our response to Reviewer #1 (Comment 1). There is currently no evidence for hippocampal-dependent impairment in these animals, but we are working to characterize this with hippocampal-dependent behavioral tasks and have demonstrated preliminary deficits in a different focal/diffuse model.

4) *Why there are no anatomical/pathological assessment of injury variability and sodium channel localization/re-localization post-injury? Can channel translocation or reassembly be considered as an alternative explanation? Were any specific post-translational modifications noted?*

These are excellent points, and the extent of Na⁺ channel translocation/reassembly/localization is an active area of investigation both in this model and in *in vitro* models of axonal injury. In this particular model, Song et. al, 2022 characterized changes in the Na_v1.6 subunit of sodium channels and compared them to human TBI cases. While this particular histopathological assessment was not performed on our pigs that underwent the electrophysiology paradigm, we expect the results to be similar given only a pig strain difference between the studies. We added this reference to the discussion section of the manuscript (page 16, lines 390-395).

Specific comments:

a) *Animal were fasted for 16 hours previously to recording. There are no data showing level of glucose in both group of animals and how this metabolic impact affects the physiological response, what is then impact of anesthesia and how this affects recordings?*

We also thought this was an important question as well when we began these experiments. We therefore tested a subset of the control and post-mTBI animals and found that levels of glucose are within normal range during our experiments. Our group has previously extensively characterized changes in physiological parameters in this model as described in detail in the references (Grovola et al., 2021; Johnson et al., 2016; Johnson et al., 2018). This references has been added to the Methods section of the manuscript (page 22, lines 578-580).

b) *In the Method Section the authors state: "The final location of the silicon probes was confirmed electrophysiologically as described previously (Ulyanova, Koch et al. 2018)." The functional methodology is described but without paralleled anatomical/stereotaxic description this consideration appeared too general and provide readers of TBI literature very limited knowledge of location with respect to injury and animal brain anatomy.*

We have extensively revised Figure 1 to give readers of TBI literature an accessible understanding of the recording and stimulation locations (page 5).

The comments C-F derive from the lack of description of methodology to provide in depth understanding of results and their impact for a reader who is not computational physiologist. The authors refer to their previous methodological papers, but they do not try describing their rationale so wide spectrum of TBI oriented audience can understand analytical details in terms of anatomy and neurophysiology.

We appreciate the goal of inclusivity in the readership for this manuscript – please see our changes below.

c) *In the Figure 1: the authors present coronal section but there is no clarity where these sections from are. 3D description of placement of stimulation and recording electrodes would be required so readers have a better knowledge of recording localization. How did authors control for similarity of electrode placement and how injury pattern was included in these considerations? What statistical analysis have been used and how cell numbers have been approximated and compared between injury models? There is always variability in the injury pattern how the reader can be assured that this variability does not impact the final conclusions.*

Thank you for your comments and suggestions. The placement of electrodes and associated anatomical error (<1mm variance) has been described in the eNeuro manuscript (Ulyanova et. al. 2018), with detailed statistical analysis of the placement based on the electrode tracks identified in the histological sections. We have also significantly edited Figure 1 to include 3D diagram of the pig brain showing electrodes' position relative to the skull bregma. Histopathological analyses of this mTBI model suggests that there is little damage directly to CA1 pyramidal neurons (see Wolf, 2017, etc.). Variability within the injured group is indeed expected, and it is captured in the statistical analysis between the control and post-TBI groups for each measure as described in the Methods section (page 24, lines 641-643; page 27, lines 722-728). To address the slightly different levels of rotation (190 rad/sec vs. 260 rad/sec), we also included the following sentence in the manuscript main text (page 8, lines 201-214) that “Interestingly, the firing rate of CA1 interneurons decreased as a rotational velocity increased, changing significantly from 7.46 ± 1.0 Hz in control animals ($n = 8$) to 3.78 ± 0.93 Hz in animals injured at ~ 190 rad/sec (188 ± 11 rad/sec, $n = 4$), and further to 2.33 ± 0.051 Hz in animals injured at ~ 260 rad/sec (256 ± 7 rad/sec, $n = 5$) ($p < 0.05$, not shown).” Our data show that even at the lowest rotational velocity (~ 190 rad/sec), electrophysiological parameters of CA1 interneurons are significantly affected by the injury (Figure R1). Please also see our responses to the Reviewer #3' questions below (page 11).

Figure R1. Firing Rate of CA1 interneurons at different levels of rotational injury (~ 190 rad/s vs. 260 rad/s).

d) *In the Figure 2: the authors compared power spectra but there are no details how and why these analyses were carried out. Were input-output curves similar? Were response variabilities assessed? Did changes in the entrainment impact the amount of information that spike rates transmitted over gamma-cycle time windows? How much output variability was unrelated to input? Why only functional assessment of the cells is presented? Can morphology of cells be investigated following the recording?*

What statistical analysis have been used and how cell numbers was approximated and compared between injury models?

Thank you for your comments. We have updated Methods to include more information on the spectral analysis and interaction between hippocampal neurons and local hippocampal oscillations and references our previous publication describing these analyses in more details (see Ulyanova et. al. 2018) (pages 6-7, lines 139-183; page 24, lines 635-649; page 27, lines 722-728).

All data shown in Figures 1, 2, and 3 have been recorded at baseline, without any stimulation applied. The main finding of this manuscript (hippocampal interneuronal dysfunction) has been observed without stimulation, and therefore, there is no input-output information added to these figures (Figure 1-3).

Only Figure 4 in this manuscript has examples of the responses to electrical stimulation, and we did not thoroughly characterize any variabilities in laminar responses to stimulation between the groups of animals (control vs. post-mTBI) in this analysis. Prior to any stimulation applied in this study (control = 3, post-mTBI = 7), we performed an input/output testing protocol and then selected a stimulation amplitude that corresponds to a half of maximum response (for each animal). On average, stimulation amplitude was in a similar range (200-500 μ A) for both control and post-mTBI groups of animals (control = 325 ± 66 vs. post-mTBI = 386 ± 54 μ A, mean \pm SEM, $p = 0.51$). Moreover, input/output curves of responses to stimulation in the Schaffer Collaterals in this pig model of concussion were previously characterized in an ex-vivo preparation (Wolf, J Neurotrauma 2017). We regret not properly describing this protocol in the first submission of the manuscript, and we now included more information on stimulation responses in the revised manuscript (page 23, lines 616-624).

The question about information in the gamma-cycle time window is intriguing – we will calculate this in the chronic awake animals, but it is likely not meaningful in the anesthetized preparation.

We have not assessed morphological changes in these neurons, but we have a manuscript in preparation for a different group of animals that should help to clarify morphological changes in neurons in a similar dataset. The number of cells detected electrophysiologically were calculated in each animal and compared between the groups using nested ANOVA (Figure R2). There were no significant differences in the number of hippocampal cells detected in control vs. post-TBI group of animals (control = 16 ± 3 cells vs. post-mTBI = 10 ± 3 cells, $p = 0.24$) (pages 4, lines 113-114).

Figure R2. Number of detected cells per animal.

e) For Figure 3, similar comments as for Figure 2.

We have updated Figure 3 accordingly. Also please see our responses to Figure 2 above.

f) In the Figure 4 some traces are presented but it is not clear if these traces are from different or same animals? This figure as it is presented seems to be descriptive and speculative. More detailed justifications would be required to match these outcomes and what is quantified in Figures 1-3. At minimum, the origin (animal and injury) of all traces must be specified.

We would like to thank the review for this comment and apologize if our description of the figure did not have enough detail to communicate our findings. Figure 4A displays averaged responses to paired pulse stimulation from the representative control (left) and post-mTBI (marked as post-mTBI₁, right) animals (page 11). However, two injured animals also displayed hyperexcitability later in the stimulation paradigm, one of which is shown in Figure 4B-D. Traces in Figure 4B are from a different injured animal (marked as post-mTBI₂) than in Figure 4A (marked as post-mTBI₁). In addition, Figures 4C, D are the zoomed version of traces displayed in Box1 and Box 2 (Figure 4B) showing epileptogenic events such as depolarizing shift, spikes, and loss of responses to stimulation (likely due to depolarization block). Control vs. post-TBI is also color-coded (blue – control, red – post-TBI).

g) *In the Figure 5 the authors apply computational model to the basket cells? Is there any experimental evidence to support that this interneuron-type has been selected? Any anatomical evidence? The authors state: “Under these modeling conditions, the window current of voltage-gated sodium channels, defined as an overlap between the activation (m_{inf}) and inactivation (h_{inf}) curves, is decreased by ~30% following inertial injury.” This is potentially important conclusion, but no direct data are offered to support this conclusion and the potential underlying mechanism.*

The reviewer is correct to state that no direct data are offered to support this conclusion, as this was a model developed to help explain the empirical data. The goal of the presented modeling studies was to narrow down the possibilities of potential causes for changes observed in hippocampal interneurons post-mTBI. This particular type of interneurons (hippocampal basket cell) was selected to assess the question for a number of reasons. As we stated previously in the discussion (lines 279-281), it has been previously shown that parvalbumin-positive (PV⁺) basket cells provide peri-somatic, feedforward inhibition in the hippocampal CA1 region (for review, see (Klausberger and Somogyi 2008; Pelkey et al. 2017)). In our previously published study, we used the same large animal model of concussion to show that input into the hippocampal CA1 region was significantly changed at 7 days post-TBI due to a loss of fiber volleys (Wolf et. al. 2017), and that there was a loss of feed-forward inhibition. Therefore, we chose the basket cell model due to their ability to stabilize hippocampal circuitry during changes in input and control the selection of cell ensembles by allowing only certain number of pyramidal cells to fire in response to the stimulus. The 30% reduction in window current (calculated as the area under the overlap of activation (m_{inf}) and inactivation (h_{inf}) curves) was calculated based on the output provided by the NEURON model for the best fit of our experimental data (shown in Figure 5). Based on these results, we will now pursue anatomical evidence for the predicted Na⁺ channel changes in the interneurons.

Reviewer #3 Modelling, hippocampal networks (Remarks to the Author):

(1) Introduction: *I could not well understand the originality of this study compared with rodents. Why have they needed to utilize minipigs as an animal model? While they briefly described unknown questions in line 83-85, how this issue is related to the use of minipigs? Please add more information for the advantages of this animal species.*

Studies utilizing rodent models of TBI can provide some mechanistic insight into electrophysiological changes following TBI (for review, see (Sandsmark et. al., 2017)). While most of the work using animal models of TBI have utilized rats and mice, rodent models of TBI are not capable of accurately representing the biomechanics of brain injury in a gyrencephalic brain, or the effects of white matter injury observed in human TBI. Gyrencephalic brain structure and grey/white matter ratios closer the human brain are important for accurate modeling of TBI (Figure R3). Due to the scaling differences (on the order of 50X larger brains in pigs) and massive increases in white matter, rat and pig models have different responses to injury. Therefore, brains of large animals such as pigs, with their gyrencephalic structure and appropriate white-to-grey matter ratios, more closely resemble human architecture and, hence, are important for an accurate biomechanical modeling of all aspects of human TBI. We have added more details to the main text of the manuscript and added a reference to our previous work (page 3, line 75-80).

Figure R3. Lissencephalic vs. Gyrencephalic Brains. Coronal sections at the level of the hippocampus in the rat, pig, and human demonstrate the increased white matter to grey matter ratio as the cortex area expands. Modeling of the white matter related biomechanics of TBI, and potentially the spread of hyperexcitability, may require a gyrencephalic animal model.

(2) *My crucial concern is their analysis after the classification of neuron types. In the Methods, they described a threshold for classification is “7 Hz”. This value itself is no problem. But they again compared firing rates of each cell type (Fig. 2 and 3) after this classification. This may be a problem because they applied statistics on a parameter after the classification based on that parameter. As shown in Figure 1, overall firing rates of all neurons are reduced by concussion. This means that some interneurons with reduced firing rates (below 7 Hz) may be misclassified as excitatory cells. If this problem happens, their results in Figure 2 (no changes in excitatory cells with a firing rate of <7 Hz) may represent just an artifact of this classification. Nonetheless, I think their results in Figure 3 are still meaningful because it demonstrates reduced firing in cell groups with a higher firing rate of >7 Hz (possibly all interneurons). As it is not easy to solve this statistical problem because the data were obtained from independent animal groups, at least, I suggest that they present all the group-comparison datasets by distributions of individual datasets (not bar graphs as they presented). In addition, as this classification is a central issue of this study, please fully described all criteria and parameters used (not just “previously published criteria” on Line 652-663).*

To address the reviewers’ insightful comments, we updated the Methods section to fully describe all criteria and parameters used in the study (page 6, lines 145-158; page 25, lines 674-685), performed a new automatic clustering algorithm to corroborate our analysis, and we include a description of the putative neuron subtype classification below.

Hippocampal CA1 cells were classified into groups of putative pyramidal cells and inhibitory interneurons (for review, see (Wheeler et al. 2015)) following previously described protocols (Broussard et al. 2020; Csicsvari et al. 1999). Briefly, the following criteria were used: 1) anatomical location: only single units from the top portion of the multichannel silicon probe were selected, which anatomically corresponded to the pyramidal CA1 cell layer; 2) firing rate threshold: putative pyramidal CA1 cells had firing rate of 7 Hz or below, while CA1 interneurons had firing rate above 7 Hz (under anesthesia); and 3) spike waveform and autocorrelogram: first moment of autocorrelogram and symmetry of spike waveforms were used to manually place cells into putative CA1 pyramidal cells vs. CA1 interneurons category similarly to previously published criteria for rodent studies (Broussard et al. 2020; Csicsvari et al. 1999).

We also performed the above automated clustering using features previously described in the rodent literature (Csicsvari et al. 1999). Namely, automated clustering of pyramidal cells and interneurons was performed using K-means clustering with $k=2$ clusters using the built-in `kmeans` function in Matlab. Neuronal subtypes were identified using the following three features: 1) firing rate, 2) spike width, and 3) first moment of the autocorrelogram. The results of automated clustering were compared to our manual clustering (Figure R4). We noted that our results from the hand clustering output were highly similar to the automated clustering. However, automated clustering moved some obvious sub-types into incorrect categories. For example, a control neuron with a firing rate of 8.3 Hz was classified as a pyramidal cell based on auto-clustering, which we believe to be incorrect. Therefore, we chose to remain with our hand-cut clusters for the final analysis.

Figure R4. Comparison of automated vs. manual clustering used for identification of putative neuronal subtypes (pyramidal vs. interneurons).

(3) *Figure 2C, D, Figure 3C, D: I think LFP power analysis on frequency bands less than 100 Hz is meaningful. However, I do not understand the meaning of “overall LFP power” on ripple-band (150-250 Hz). Generally, we directly detect transient “ripple events” based on RMS or Hilbert transformation, which will be more useful to discuss how memory consolidation and retrieval mechanisms are altered. I suggest that they employ similar ripple analyses to previous studies (e.g., the frequency of ripple events, amplitude of ripple events, and ripple-triggered spike rates).*

We would like to thank the reviewer for their questions about overall LFP power and ripple band power in particular. In general, we do not observe sharp-wave ripple events under isoflurane anesthesia in our pig *in-vivo* experiments. As we mentioned in our rodent injury model study (Koch et al. 2020), high-frequency oscillations recorded within CA1 pyramidal layer likely reflect local processes, whether they are coordinated spiking activity of local neurons or synaptic activity between pyramidal cells and local interneuronal networks as occurs in ripple events (also see (Schomburg et al. 2012)). High-frequency power losses after injury may indicate a loss of overall local neuron firing (whether because of reduced firing rates or reduced neuron numbers), reduced synchrony of neuronal populations, or a loss of pyramidal cell–interneuron interactions. For more details on the analysis of the “overall LFP power” on ripple-band (150-250 Hz) under isoflurane anesthesia, see (Koch et al. 2020) for more details. We have demonstrated ripples in the awake-behaving pig (Ulyanova, 2019) and are working to characterize their differences in our pig TBI models. We agree it is an interesting area in relation to cognitive dysfunction.

(4) *Figure 4: These graphs and related sentences have no quantification. Please quantify these data and describe their claims based on statistical results. For example, fEPSP amplitude (Fig. 4A), duration or area of sustained depolarization (Fig. 4B), and the number of unit signals (Fig. 4C) could be quantified. By the way, why they applied paired pulse stimulation? Generally, this stimulation is utilized*

to analyze presynaptic release. Please describe changes in paired pulse ratio to discuss synaptic changes.

Thank you for your question. Please see our responses to Reviewer #2 above (Specific Questions, section d, page 7).

(5) Line 614-617: They used two types of silicon probes (NN32/LIN and NN32/TET). To my experience, NN32/LIN is useful to capture multiunit signals? (As recording points are linearly located with a large interval). Please clarify how many cells or how many LFP signals were captured by these two types of how many probes in the Methods or Results part.

We thank the reviewer for their comments regarding two types of the electrodes used in the study. In this study, NN32/LIN probes were used to record laminar LFPs (32 channels each) in 9 animals (control = 5 vs. post-mTBI = 4), and NN32/TET probes were used to record LFPs in 8 animals (control = 3 vs. post-mTBI = 5). We added this information to the Methods section of the manuscript (page 23, lines 602-6605; page 24, lines 639-649).

Over the years, our group used both types of these electrodes (in linear configuration vs. with tetrodes) for acute as well as chronic in vivo recordings (see (Ulyanova et al. 2019; Ulyanova et al. 2018)). The spike sorting software used to separate single units from the multi-unit activity works well with both types of these probes, and we have not noted any significant differences in the number of single units recorded with either probe. Total number of single units recorded in control ($n_{\text{cells}} = 125$) and post-TBI ($n_{\text{cells}} = 93$) groups were used for comparison between the groups (page 4, lines 112-117), and the number of recorded cells per animal was not different between the groups (control = 16 ± 3 cells vs. post-mTBI = 10 ± 3 cells, mean \pm SEM, $p = 0.24$). Additionally, we found no significant differences between a number of single units recorded with two types of electrodes (NN32/TET = 12 ± 3 cells vs. NN32/LIN = 14 ± 3 cells per animal, mean \pm SEM, $p = 0.65$, Figure R5) or between a number of single units recorded with each type of electrode for control vs. post-mTBI groups of animals (NN32/TET: control = 16 ± 6 vs. post-mTBI = 9 ± 3 cells per animal, mean \pm SEM, $p = 0.26$; and NN32/LIN: control = 15 ± 5 vs. post-mTBI = 12 ± 6 cells per animal, mean \pm SEM, $p = 0.69$). We added this information to the Methods section of the manuscript (page 24, lines 638-649).

NN32/TET vs NN32/LIN

Figure R5. Number of single units recorded with NN32/TET vs. NN32/LIN probes.

References

- Broussard, John I., John B. Redell, Jing Zhao, Mark E. Maynard, Nobuhide Kobori, Alec Perez, Kimberly N. Hood, Xu O. Zhang, Anthony N. Moore, and Pramod K. Dash. 2020. 'Mild Traumatic Brain Injury Decreases Spatial Information Content and Reduces Place Field Stability of Hippocampal CA1 Neurons', *Journal of Neurotrauma*, 37: 227-35.
- Buzsáki, György. 2015. 'Hippocampal sharp wave-ripple: A cognitive biomarker for episodic memory and planning', *Hippocampus*, 25: 1073-188.
- Csicsvari, Jozsef, Hajime Hirase, András Czurkó, Akira Mamiya, and György Buzsáki. 1999. 'Oscillatory Coupling of Hippocampal Pyramidal Cells and Interneurons in the Behaving Rat', *The Journal of Neuroscience*, 19: 274.
- Grovala, Mr Auid-Orcid, N. Paleologos, D. P. Brown, N. Tran, K. L. Wofford, J. P. Harris, K. D. Browne, P. A. Shewokis, J. A. Wolf, Dk Auid-Orcid Cullen, and Je Auid-Orcid Duda. 2021. 'Diverse changes in microglia morphology and axonal pathology during the course of 1 year after mild traumatic brain injury in pigs', *Brain Pathol.*, 31(5): e12953.
- Klausberger, Thomas, and Peter Somogyi. 2008. 'Neuronal Diversity and Temporal Dynamics: The Unity of Hippocampal Circuit Operations', *Science*, 321: 53.
- Koch, Paul F., Carlo Cottone, Christopher D. Adam, Alexandra V. Ulyanova, Robin J. Russo, Maura T. Weber, John D. Arena, Victoria E. Johnson, and John A. Wolf. 2020. 'Traumatic Brain Injury Preserves Firing Rates but Disrupts Laminar Oscillatory Coupling and Neuronal Entrainment in Hippocampal CA1', *eNeuro*: ENEURO.0495-19.2020.
- Pelkey, Kenneth A., Ramesh Chittajallu, Michael T. Craig, Ludovic Tricoire, Jason C. Wester, and Chris J. McBain. 2017. 'Hippocampal GABAergic Inhibitory Interneurons', *Physiological Reviews*, 97: 1619-747.
- Schomburg, Erik W., Antonio Fernández-Ruiz, Kenji Mizuseki, Antal Berényi, Costas A. Anastassiou, Christof Koch, and György Buzsáki. 2014. 'Theta Phase Segregation of Input-Specific Gamma Patterns in Entorhinal-Hippocampal Networks', *Neuron*, 84: 470-85.
- Schomburg, Erik W., Costas A. Anastassiou, György Buzsáki, and Christof Koch. 2012. 'The Spiking Component of Oscillatory Extracellular Potentials in the Rat Hippocampus', *The Journal of Neuroscience*, 32: 11798.
- Ulyanova, Alexandra V., Carlo Cottone, Christopher D. Adam, Kimberly G. Gagnon, D. Kacy Cullen, Tahl Holtzman, Brian G. Jamieson, Paul F. Koch, H. Isaac Chen, Victoria E. Johnson, and John A. Wolf. 2019. 'Multichannel Silicon Probes for Awake Hippocampal Recordings in Large Animals', *Frontiers in Neuroscience*, 13: 397.
- Ulyanova, Alexandra V., Paul F. Koch, Carlo Cottone, Michael R. Grovala, Christopher D. Adam, Kevin D. Browne, Maura T. Weber, Robin J. Russo, Kimberly G. Gagnon, Douglas H. Smith, H. Isaac Chen, Victoria E. Johnson, D. Kacy Cullen, and John A. Wolf. 2018. 'Electrophysiological Signature Reveals Laminar Structure of the Porcine Hippocampus', *eNeuro*, 5: ENEURO.0102-18.2018.
- Wheeler, Diek W., Charise M. White, Christopher L. Rees, Alexander O. Komendantov, David J. Hamilton, and Giorgio A. Ascoli. 2015. 'Hippocampome.org: a knowledge base of neuron types in the rodent hippocampus', *eLife*, 4: e09960.

Reviewer #1 (Remarks to the Author):

All my concerns have been addressed.

Reviewer #2 (Remarks to the Author):

Thank you for responding to most of my concerns, please see below remaining comments (related to my previous review) that would address remaining limitations of the authors approach.

1) The in vivo finding should provide expanded/broader evidence for remodeling in hippocampal network, this should include both electrical activity and putative contribution from vascular supplies. There are no data showing that vascular/oxygen supply to hippocampus is not altered by injury.

The authors response is based on their concerns related to BBB. However altered blood supply related to vascular remodeling is still not addressed or considered in their reply. In the Discussion authors should address this concern as well.

2) Does anesthesia and its level impact the electrophysiological activity in the similar manner in control and injured animals?

The authors state: "We don't believe that a specific impact of the anesthesia on injured interneurons is likely. We are currently working on the awake behaving pig electrophysiology study, and we hope to be able to answer these questions in the future."

I appreciate this reply but in this current manuscript in the Discussion author should address this potential concern by presenting some evidence related to reasons behind this believe.

3) What is behavioral evidence for hippocampal impairments?

Please see our response to Reviewer #1 (Comment 1). There is currently no evidence for hippocampal dependent impairment in these animals, but we are working to characterize this with hippocampal dependent behavioral tasks and have demonstrated preliminary deficits in a different focal/diffuse model.

I agree with authors that they can speculate about hippocampus role in chronic post-TBI neurological impairments, however the question remain if this primary impacted region. It seem that authors suggest that independently on injury mechanism this may be the case. In the Discussion please explain why?

3) Animal were fasted for 16 hours previously to recording. There are no data showing level of glucose in both group of animals and how this metabolic impact affects the physiological response, what is then impact of anesthesia and how this affects recordings?

The authors replied: "We also thought this was an important question as well when we began these experiments. We therefore tested a subset of the control and post-mTBI animals and found that levels of glucose are within normal range during our experiments. Our group has previously extensively characterized changes in physiological parameters in this model as described in detail in the references (Grovola et al., 2021; Johnson et al., 2016; Johnson et al., 2018). This references has been added to the Methods section of the manuscript (page 22, lines 578-580)."

I appreciate that authors added this sentence but the concern related to glucose level has not been directly addressed in this sentence. This would be critical in the context of the works done in Ryan's

laboratory at Cornell Medical College, New York and its implication for authors' simulation model, conclusions and interpretation (e.g. Sci Adv. 2021 Dec 3;7(49):eabi9027. doi: 10.1126/sciadv.abi9027. Epub 2021 Dec 3.

Synaptic vesicle pools are a major hidden resting metabolic burden of nerve terminals and his comprehensive review: "The control of release probability at nerve terminals." PMID: 30647451), which also relates to point 1 (see above) and concern expressed by 3rd reviewer (comment 4). This is not about revisiting the proposed model but about clearly stated limitation of the approach.

Reviewer #3 (Remarks to the Author):

The authors have adequately addressed my concerns. I have no further major comments.
(Minor)

- Please confirm font size in all graphs. For example, the letters near the scale bars in Figure 1 are too small, compared with Figure 4. This might be something to consider for all graphs, similarly.

Title: "Hippocampal Interneuronal Dysfunction and Hyperexcitability in a Porcine Model of Concussion"

Manuscript: COMMSBIO-22-0908A

Response to Reviews:

We are grateful for the thorough reviews that were provided to us on our manuscript, as well as the detailed suggestions for improvement. We believe we have been able to address the reviewers' and editors' suggestions in this final revision.

Reviewer #2 (Remarks to the Author):

We have incorporated all Reviewer #2 suggestions into Discussion section of the main text as a part of the study's limitations (see below, also page 20, lines 516-530 of the main text):

There are several limitations to our electrophysiological approach, including the anesthetized preparation. The predominant effects of isoflurane are known to be on GABA, glycine, and NMDA receptors, and we cannot rule out differential effects on interneurons vs. pyramidal cells (Jones et al., 1992; Qiu et al., 2023). However, we have no evidence to suggest that anesthesia would differentially affect intrinsic properties or synaptic release probabilities of *injured* interneurons vs. pyramidal neurons post-injury (Dittman and Ryan, 2019; Pulido and Ryan, 2021). Differences in metabolic states may also contribute to the observed electrophysiological changes, along with potential injury-induced vascular remodeling or changes in neurovascular coupling (Salehi et al., 2017). We selectively focused on hippocampal area CA1 due to our prior results demonstrating profound changes in this region in a hippocampal slice preparation with this model, as well the known role of the hippocampus in various aspects of cognition and its selective vulnerability in TBI and PTE. Verification of these data in awake, chronically implanted animals and expansion to other limbic areas involved in these processes is important future work (Danielli et al., 2023).

We also updated the references section to include additional citations suggested by the Reviewer #2. Below is our point-by-point response to the reviewers' comments:

- 1) *The in vivo finding should provide expanded/broader evidence for remodeling in hippocampal network, this should include both electrical activity and putative contribution from vascular supplies. There are no data showing that vascular/oxygen supply to hippocampus is not altered by injury. The authors' response is based on their concerns related to BBB. However, altered blood supply related to vascular remodeling is still not addressed or considered in their reply. In the Discussion authors should address this concern as well.*

Thank you for pointing this out to us. We appreciate Reviewer's comment. We agree that one of the limitations to our approach is that it does not detect other potential contributors to these electrophysiological changes such as vascular remodeling or other changes in neurovascular coupling. We have incorporated Reviewer #2's comment into Discussion section of the manuscript.

2) *Does anesthesia and its level impact the electrophysiological activity in the similar manner in control and injured animals? The authors state: “We don’t believe that a specific impact of the anesthesia on injured interneurons is likely. We are currently working on the awake behaving pig electrophysiology study, and we hope to be able to answer these questions in the future.” I appreciate this reply but in this current manuscript in the Discussion author should address this potential concern by presenting some evidence related to reasons behind this believe.*

Another limitation to our study is that it was performed under the inhaled anesthetic, isoflurane, which acts on multiple receptors and channels (i.e., GABA, glycine, and NMDA). We have no data to suggest this would differentially affect intrinsic properties of interneurons vs. pyramidal neurons, but verification of these data in awake animals will be important future work.

3) *What is behavioral evidence for hippocampal impairments? There is currently no evidence for hippocampal dependent impairment in these animals, but we are working to characterize this with hippocampal dependent behavioral tasks and have demonstrated preliminary deficits in a different focal/diffuse model. I agree with authors that they can speculate about hippocampus role in chronic post-TBI neurological impairments, however the question remain if this is the primary impacted region. It seems that authors suggest that independently on injury mechanism this may be the case. In the Discussion, please explain why?*

Thank you for your questions. We agree with the Reviewer #2 that other brain regions may be affected by the injury, and it is important to discuss this in the manuscript. In this study, we selectively focused on hippocampal area CA1 due to our prior results demonstrating profound changes in this region in a hippocampal slice preparation with this model, as well the known role of the hippocampus in various aspects of cognition and its selective vulnerability in traumatic brain injury and post-traumatic epilepsy. Verification of these data in awake, chronically implanted animals and expansion to other limbic areas involved in these processes is important future work.

4) *Animals were fasted for 16 hours previously to recording. There are no data showing level of glucose in both group of animals and how this metabolic impact affects the physiological response, what is then impact of anesthesia and how this affects recordings? I appreciate that authors added this sentence, but the concern related to glucose level has not been directly addressed in this sentence. This would be critical in the context of the works done in Ryan’s laboratory at Cornell Medical College, New York and its implication for authors’ simulation model, conclusions, and interpretation (e.g., Sci Adv. 2021 Dec 3;7(49):eabi9027. doi: 10.1126/sciadv.abi9027. Epub 2021 Dec 3. Synaptic vesicle pools are a major hidden resting metabolic burden of nerve terminals and his comprehensive review: “The control of release probability at nerve terminals.” PMID: 30647451), which also relates to point 1 (see above) and concern expressed by 3rd reviewer (comment 4). This is not about revisiting the proposed model but about clearly stated limitation of the approach.*

Thank you for providing these redirecting our attention to these references. We do agree that synaptic release probability may be an important contributor to the observation of this study, and therefore we added the suggested references as well as described this concept as another limitation in the Discussion section of the manuscript.

Reviewer #3 (Remarks to the Author):

Thank you for noticing that font sizes are different in the Figures. We appreciate your attention to details and have revised all figures accordingly to your and editors’ suggestions.